# *Corex*: Pushing the Boundaries of Complex Reasoning through Multi-Model Collaboration

**Qiushi Sun**$^{\diamond\heartsuit*}$**Zhangyue Yin**$^{\clubsuit}$ **Xiang Li**$^{\spadesuit}$ **Zhiyong Wu**$^{\diamond\dagger}$ **Xipeng Qiu**$^{\clubsuit}$ **Lingpeng Kong**$^{\heartsuit}$
$^{\diamond}$Shanghai AI Laboratory $^{\heartsuit}$The University of Hong Kong
$^{\clubsuit}$Fudan University $^{\spadesuit}$East China Normal University
qiushisun@connect.hku.hk, yinzy21@m.fudan.edu.cn, xiangli@dase.ecnu.edu.cn
wuzhiyong@pjlab.org.cn, xpqiu@fudan.edu.cn, lpk@cs.hku.hk

## Abstract

Large Language Models (LLMs) are evolving at an unprecedented pace and have exhibited considerable capability in the realm of natural language processing (NLP) with world knowledge. Benefiting from ultra-large-scale training corpora, a single LLM can manage typical NLP tasks competently. However, its performance in executing complex tasks is still confined by the limitations of its internal representation. To push this boundary further, we introduce *Corex*, a suite of novel general-purpose strategies that transform LLMs into autonomous agents, pioneering multi-model collaborations for task-solving. Inspired by human behaviors, *Corex* is constituted by diverse collaboration modes including Discuss, Review, and Retrieve, which collectively work towards enhancing the reasoning process. These paradigms foster task-agnostic approaches that enable LLM-based agents to "think outside the box," thereby overcoming common errors and providing better solutions. Through extensive experiments across four different types of reasoning tasks, we demonstrate that orchestrating multiple LLM-based agents to work in concert yields better results compared to well-established existing baselines. Further analysis reveals the advantages of *Corex* over other multi-model methods, synergies produced among different LLMs, and the effectiveness across various aspects. Codes and data are available at https://github.com/QiushiSun/Corex.

## 1 Introduction

Large Language Models (LLMs) have succeeded in advancing the state-of-the-arts for a series of Natural Language Processing (NLP) tasks (Brown et al., 2020; Chowdhery et al., 2022; OpenAI, 2023; Touvron et al., 2023; Zhao et al., 2023a, *inter alia*). Recent research (Wei et al., 2022a) indicates that scaling up models (Kaplan et al., 2020) can yield improvements in both performance and sample efficiency across a broad spectrum of downstream tasks. Notwithstanding their remarkable proficiency in language understanding and instruction following (Ouyang et al., 2022), the reasoning abilities of LLMs, often seen as a hallmark for assessing their potential, still present challenges (Suzgun et al., 2023; Huang & Chang, 2023). Concurrently, there is a prevailing view that merely increasing the size might not adequately address their inherent limitations in solving reasoning tasks (Rae et al., 2022).

In response to this challenge, Wei et al. (2022b) put forth chain-of-thought (CoT) prompting that an LLM generates a series of intermediate steps toward a final answer, contrasting the use of "answer-only" prompts. Subsequently, various approaches have been put forward, such as self-consistency decoding (Wang et al., 2023d) which utilizes a majority voting mechanism to determine the final answer, and program-aided language models (PAL; Gao et al., 2022; Chen et al., 2022a) that leverage code generation to reduce errors in

---

* Work done during an internship at Shanghai AI Laboratory.
$^{\dagger}$ Corresponding Author.

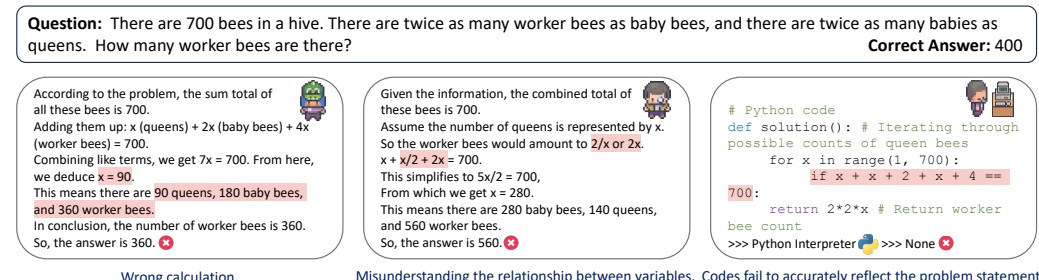

Figure 1: A depiction of three prevalent errors observed across LLMs when employing CoT and PAL to conduct reasoning tasks.

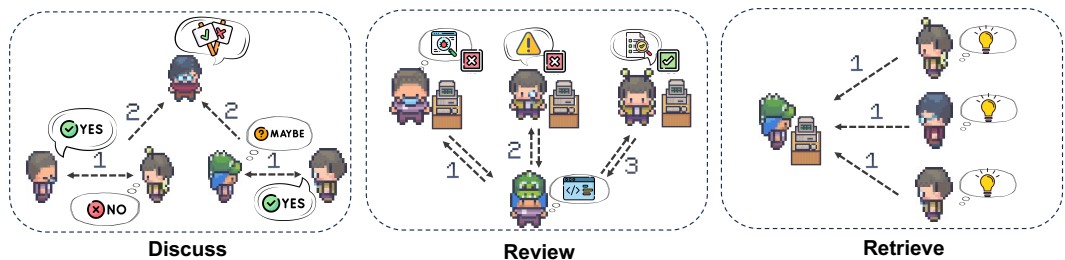

Figure 2: An intuitive illustration of *Corex*, employs LLM-based agents to solve a problem collaboratively. The strategies encompass Discuss, Review, and Retrieve modes, leveraging both the reasoning process and code synthesis. Numerals represent the order of the process.

computations. Besides, curated prompts necessitate task-specific designs (Zheng et al., 2023a) have also been utilized to elicit more accurate predictions. Nevertheless, these approaches are confined within a static black box (Yao et al., 2023b), wherein the LLM relies exclusively on its internal representation for generating responses and is prone to generating unreliable answers (Ji et al., 2023; Yin et al., 2023b). These shortcomings underscore that relying solely on crafting decoding strategies and specialized prompts may not serve as a silver bullet for addressing complex reasoning tasks (Qiao et al., 2023). Moreover, recent studies (Huang et al., 2023; Valmeekam et al., 2023; Stechly et al., 2023) reveal that LLMs struggle to reach better responses through self-correction (Madaan et al., 2023). Alternatively, transforming LLMs into autonomous agents and enabling them to "think outside the box" emerges as a promising yet underexplored pathway.

Within the realm of well-established sociological concepts, multiple cognitive processes interact and cooperate will produce a combined effect that is greater than the sum of their individual contributions (Luppi et al., 2022). This principle is echoed within artificial intelligence (Li et al., 2023a). Although the study of intelligent agents has been explored for decades (Minsky, 1988; 2007), the advent of LLMs has rejuvenated interest in building language agents (Sumers et al., 2024) and introduced more challenges. An emerging perspective is that encouraging collaboration and communication between models could potentially pave the way for a new stage for enhancing complex reasoning capabilities.

In this study, we propose *Corex*, a suite of human-inspired strategies that leverages multi-model collaboration to elicit reasoning for complex task-solving. To facilitate synergies between models, we first assign distinct personas to different LLM-based agents, followed by the design of various collaborative paradigms. This collective intelligence-based method aims to conquer prevalent obstacles in the current landscape of reasoning, as exemplified in Figure 1. It also endeavors to alleviate common issues observed in majority voting-based methods, where accurate responses might be overwhelmed by incorrect ones and exorbitant costs. To be specific, *Corex* configures LLMs as a group of autonomous agents, adopting the paradigms shown in Figure 2 for multi-model collaboration: (1) *Discuss*, utilizing group-based discussions among models to effectively enhance factuality and diversity of rationales, thereby reducing fallacies and hallucinations; (2) *Review*, enabling models to scrutinize reasoning chains or synthesized codes from their counterparts to ensure the correctness of

generated contents, coupled with potential refinements; (3) *Retrieve*, aiming to enable the model to identify the most faithful option from a pool of candidate chains by scoring and ranking, facilitates a higher degree of alignment with the final response. Together, *Discuss*'s critical analysis, *Review*'s focus on reliability, and *Retrieve*'s alignment with faithful response - create a comprehensive approach to address various complex reasoning tasks.

We conduct extensive experiments across four types of tasks: mathematical reasoning, symbolic reasoning, commonsense reasoning, and semi-structured reasoning. The results illustrate that our method achieves substantial performance gains over previous strong baselines. Moreover, each mode distinctly excels in different categories of tasks, showcasing its specific strengths. Further exploration reveals *Corex*'s advantages over existing multi-model approaches and synergies can occur between different modes/LLM backbones. Additionally, *Corex* reduces the reasoning overheads, achieving multifaceted cost-effectiveness.

## 2 Related works

**Chain-of-Thought Prompting Elicits LLM Reasoning.** Chain-of-Thought (CoT; Wei et al., 2022b) prompting, as one of the celebrated capabilities of recent LLMs, is a pivotal breakthrough for performing complex multi-step reasoning when provided with limited examples. Further variants show that CoT can be improved by adding certain "magic phrases" (Kojima et al., 2022), automated demonstrations construction (Zhang et al., 2023a), reasoning in different modalities (Zhang et al., 2023b; Yang et al., 2023; Yao et al., 2023c), and applying modular approaches (Khot et al., 2023). For robustness, researchers transform problems into interleaved reasoning chains (Zhou et al., 2023; Lyu et al., 2023) or adopt ensembling (Wang et al., 2022). Notably, self-consistency (Wang et al., 2023d) select answers from multiple reasoning paths by frequency, have greatly elevated the performance in complex reasoning. This approach has been further optimized by utilizing prompts with higher complexity (Fu et al., 2023c). Lately, Yao et al. (2023a) employ heuristic-guided search on "trees" constructed from thoughts to assist LLMs in navigating the problem space. Tool utilization (Qin et al., 2023) represents another pathway to enhance reasoning, which are commonly used for addressing numerical problems. Representatives are Program-aided Language model (PAL; Gao et al., 2022) or Program of Thoughts (PoT; Chen et al., 2022a). Such an approach utilizes LLMs to interpret NL problems, generating programs as intermediate reasoning steps that will be offloaded to a Python interpreter for execution to get final solutions (Ni et al., 2023).

**Multi-Model Synergy for Task Solving.** Utilizing multiple LLMs collectively to solve problems is still in its preliminary stages. The cornerstone of collaboration is constructing a human-like reasoning architecture (Zhu et al., 2023) for LLMs under different environments (Liu et al., 2023). Fu et al. (2023b) investigate whether multiple LLMs can autonomously enhance their performance through mutual interactions. Du et al. (2023) and Liang et al. (2023) explore enhancing the factuality of specific tasks, e.g., translation and arithmetic reasoning, by facilitating "debates" among multiple models. LLMs' collaboration has also been applied to software development (Qian et al., 2023) and text evaluation (Chan et al., 2023) by assigning identities to models to simulate the development process. Furthermore, from the perspective of social intelligence, inducing cognitive synergy and having them take on different characters (Wang et al., 2023e) during task execution has been proven to have significant potential (Sclar et al., 2023). Recent explorations into artificial societies (Park et al., 2023) aim to leverage collective intelligence to emulate the efficiency of human social structures (Li et al., 2023a; Webb et al., 2023). More recently, Yin et al. (2023a) proposed cross-model commutation based on network topological structures.

## 3 *Corex*

We propose *Corex*, a suite of collaborative reasoning approaches inspired by human social interactions. Three main components: Discuss, Review, and Retrieve modes are introduced in the following sections. Let us assume a set of LLM-based agents $\{A_1, A_2, \ldots, A_n\}$ participating in multi-model collaboration. Each agent $A_i$ generates the corresponding reasoning chain $c_i$ and its prediction $p_i$ when facing a query $q$.

### 3.1 Discuss

Information exchange between models contributes to enhanced factuality (Wang et al., 2023a). In Discuss mode, our LLM-based agents are divided randomly into two groups, the Blue Team and the Green Team, with one reserved as a judge denoted as $A_j$. The discussion within one team involves several rounds, limited to a maximum of $T$ rounds of communications. In each round $t$ ($t = 1, 2, \ldots, T$), the agents engage in iterative discussions[1] to refine their reasoning chains and predictions. This dynamic interaction allows for the continual modification of viewpoints for $q$, as expressed by $c_i^t$ and predictions $p_i^t$.

Each team then presents their refined predictions $p_{\text{blue}}^t$ and $p_{\text{green}}^t$ at the end of each round. If both teams consistently agree throughout the interactions, i.e., $p_{\text{blue}}^t = p_{\text{green}}^t$, the discussion concludes smoothly. However, in the instance of a discrepancy between the teams' predictions, every output from each round is presented to $A_j$. The judge employs a decision-making process $h$ to evaluate the quality of the reasoning chains and predictions from each round of the discussion. The conclusion is determined by $h(c_{\text{blue}}^t, p_{\text{blue}}^t, c_{\text{green}}^t, p_{\text{green}}^t)$ across all rounds, ensuring a comprehensive assessment and a more informed final decision.

Diverging from previous works (Du et al., 2023; Liang et al., 2023; Xiong et al., 2023) employing "debates" among LLMs, our approach adopts the concept of group discussions. We opt not to facilitate models in jointly exchanging their reasoning processes to converge on a single common answer for several reasons: (1) The limited context length inhibits the ability to hold the communication process, (2) A single final answer is not always correct, as erroneous consensus or biases among models can occur (Wang et al., 2023c), (3) Given performance gaps among various LLMs, there is a risk of strong ones "monopolizing" the collaborations, thereby overshadowing the insights from others. Therefore, we aim to preserve both the factuality and the diversity of thoughts among players.

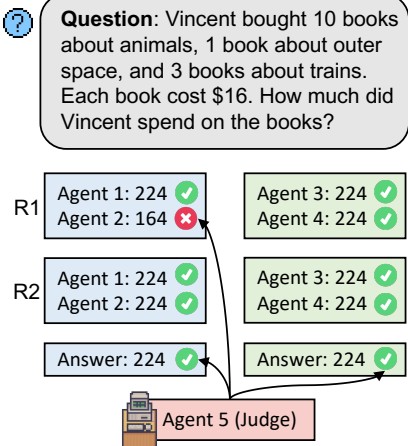

Figure 3: Illustration of 2 rounds of discussion, rationales omitted.

### 3.2 Review

Within the scope of reasoning, both CoT and PAL are effective methods with distinct strengths. Grounded in natural language, CoT-based methods stand out for the generality and the clarity of explanations. In contrast, facilitated by programs, PAL guarantees computational accuracy (Zhao et al., 2023b). However, they both exhibit drawbacks due to the reliance on LLMs' internal representations. For CoT and its variants, issues are twofold: (1) Cumulative errors, where mistakes tend to amplify and propagate throughout the reasoning chain; and (2) A plateau in text quality that cannot be substantially improved through prompting (Xu et al., 2022; Li et al., 2023b). Alternatively, PAL faces its own challenges: (1) LLMs might misinterpret questions, which inadvertently results in technically correct yet misguided programs; and (2) Generated codes are not always error-free: LLMs may potentially write buggy codes, such as referencing undefined variables or engaging in "Division by Zero" operations. Inspired by recent efforts of LLMs peer-rating (Zheng et al., 2023b) and collaborative coding practices prevalent in software engineering, we introduce the Review mode to address the aforementioned issues through multi-agent collaboration. To be specific, a single agent $A_p$ is randomly selected to act as the primary agent. Initially, $A_p$ takes the responsibility of formulating corresponding reasoning chains for $q$ along with the prediction, and crafting codes if required. This

---

[1] Due to the context length limit of `GPT-3.5-Turbo`, only information from the previous round is stored during the discussion.

initial collection of solutions is represented as $S_p^{(0)} = \{a_p, c_p, m_p\}$, where $a_p$, $c_p$, and $m_p$ signify the answer, reasoning chain, and codes respectively. $S_p^{(0)}$ is then subjected to iterative reviews by the other agents that function as reviewers in a sequential manner, rigorously scrutinizing both the reasoning chain and the code formulated by $A_p$ or modified by preceding reviewers. It is crucial to highlight that each reviewer receives input from its predecessors, signifying that each subsequent review is grounded on the outcomes and feedback of the preceding ones, fostering a progressively refined solution.

The reviewing process is formalized as $S_p^{(i+1)} = R_i(S_p^{(i)}, F_i)$, where $R_i$ encapsulates the review outcome at the $i^{th}$ iteration and $F_i$ represents the feedback received. In essence, the solution set $S_p^{(i+1)}$ results from an enhancement of its preceding version $S_p^{(i)}$, informed by the feedback $F_i$. Following the completion of all review iterations, the outcome is determined by the final iteration of the solution set $S_p^{(n-1)}$. Specifically, the final prediction $a_p^{(n-1)}$ is chosen as the answer for $q$, and in instances where code is involved, the last revised version $m_p^{(n-1)}$ is executed by a Python interpreter to produce the outcome.

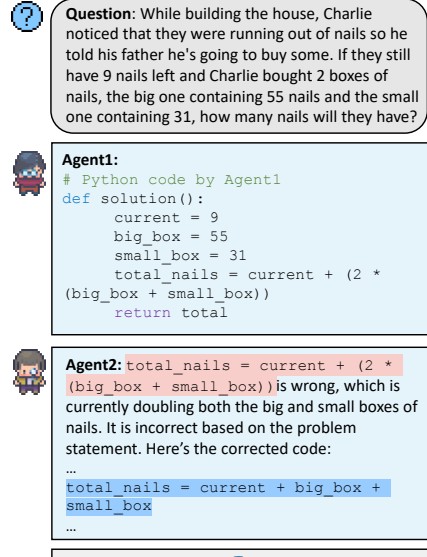

Figure 4: Illustration of reviewing code generated by other agents (1st round).

## 3.3 Retrieve

In the final thread of work, we delve into the Retrieve mode to identify the most faithful answer through collaborations. While previous strategies based on majority voting mechanism (Wang et al., 2023d; Fu et al., 2023c) can mitigate the low-diversity issue of techniques such as beam-search (Li & Jurafsky, 2016), they still present the following two significant challenges:

(1) Correct answers risk being swayed by incorrect ones. (2) Despite facilitating a notable enhancement in performance, it exponentially escalates the computational burden and tends to reach a performance "saturation point" as the sampled chains increase. We attribute these drawbacks to the limited scope of majority voting techniques that singularly prioritize the prediction while overlooking the faithfulness of reasoning chains (Li et al., 2023c). In response, we propose the Retrieve mode, a paradigm specifically engineered to evaluate whether the answer can be expressed by the content (explanation) generated during reasoning (Jacovi & Goldberg, 2020; Lanham et al., 2023).

Concretely, given a query $q$, we randomly select an agent $A_r$ from the pool of $n$ agents to act as the retriever. The remaining agents $\{A_1, A_2, \ldots, A_{n-1}\}$ independently solve $q$. Each of these agents de-

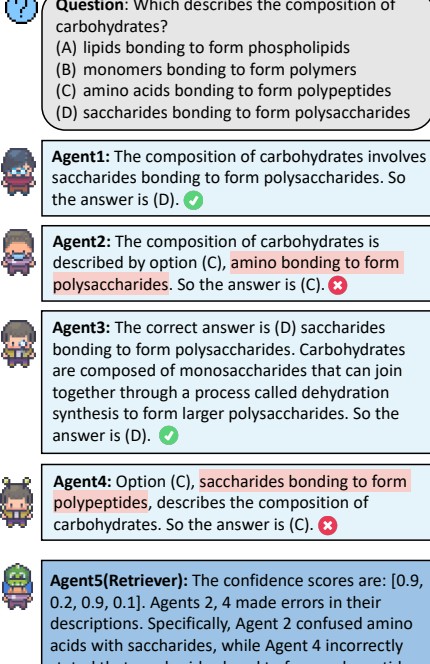

Figure 5: Illustration of retrieving reasoning chains with their answers.

rives its own reasoning chains $c_i$ and corresponding predictions $p_i$. Together, they form a candidate pool, denoted by $\mathcal{P} = \{(c_i, p_i)\}_{i=1}^{n-1}$.

The retriever $A_r$ then scrutinizes the candidates in $\mathcal{P}$. For $(c_i, p_i)$, $A_r$ evaluates the faithfulness between $c_i$ and $p_i$. Based on this assessment, the retriever assigns a confidence score $s_i$ in the range $[0, 1]$, which is denoted as: $s_i = f_r(c_i, p_i)$ where $f_r$ indicates the retriever's evaluation process. After that, the most faithful response to the question $q$ is then determined by the highest confidence: $(c^*, p^*) = \operatorname{argmax}_{(c_i, p_i) \in \mathcal{P}} s_i$. Here, $(c^*, p^*)$ denotes the chain-prediction pair that the retriever considers most faithful, which will serve as the final answer for the query $q$.

Retrieve mode enables the selection of the most aligned combination of reasoning chains and answers from a diversified candidate pool. Distinct from previous text quality assessment methods, which rely on the log probability of sequences (Adiwardana et al., 2020) that is computationally inefficient and often unavailable for commercial LLMs, our approach is entirely predicated on model-to-model interactions (Chen et al., 2023b) and is reference-free.

## 4 Experiment

### 4.1 Experimental Setup

**Tasks and Datasets.** We evaluate *Corex* across four types of tasks: (1) Arithmetic reasoning across various difficulties, such as GSM8K (Cobbe et al., 2021) and GSM-Hard (Gao et al., 2022). (2) Commonsense reasoning covering four datasets, exemplified by StrategyQA (Geva et al., 2021). (3) Symbolic reasoning incorporating four tasks derived from BigBench (bench authors, 2023). (4) Semi-structured understanding, like FinQA and ConvFinQA (Chen et al., 2021b; 2022b). Detailed descriptions and statistics of tasks are listed in Appendix E.

**Baselines.** We compare *Corex* with several widely used strong baselines. (1) Chain-of-Thought prompting (CoT; Wei et al., 2022b). (2) Self-Consistency (CoT-SC; Wang et al., 2023d), which employs a majority voting mechanism to select the most consistent answer from several reasoning chains as the final answer. (3) Complexity-based consistency (ComplexCoT; Fu et al., 2023c) that selects the majority answer from the candidates with higher reasoning complexity. (4) Program-aided language model (PAL/PoT; Gao et al., 2022; Chen et al., 2022a) that uses LLMs to generate programs as intermediate reasoning steps, while offloading the computation to a Python interpreter. For simplicity, we denote CoT-SC(x) and ComplexCoT(x) in our experiments and analysis to represent cases utilizing different reasoning paths, where "x" indicates the number of output chains. For all baselines, we adhere to the few-shot exemplars to ensure fair comparisons, detailed in Appendix B.

**Implementation Details.** We access OpenAI and Anthropic models through their respective APIs. Specifically, we employ GPT-3.5-Turbo-0613 for evaluating both *Corex* and baseline methods in the main experiments. Moreover, in further experiments and analysis involving different LLMs for collaboration, we also incorporate the use of GPT-4-0613, Claude-Instant-1.2, and LLaMA-2-Chat(7B/13B). The details of prompts and hyperparameter settings for both baselines and *Corex* are in Appendix G.

### 4.2 Main Results

We report the results of *Corex* over 18 tasks across four categories. For each kind of task, the best results are highlighted in **bold** and the second best results are marked with underline. For Review mode, we use *Corex-Review_{NL}* and *Corex-Review_{Code}* to describe the scenarios that use NL reasoning chains or codes as the intermediate steps, respectively. All modes within *Corex* are configured to operate with 5 LLM-based agents, ensuring favorable cost-effectiveness. For *Corex-Discuss*, the upper bound of discussion rounds is set to 5.

**Mathematical Reasoning.** Table 1 shows the results across arithmetic tasks with varying difficulties. Our method achieves notable performance improvements on most benchmarks.

Table 1: Comparison of accuracy on seven mathematical reasoning datasets using various *Corex* modes and strong baselines.

|  | MultiArith | SVAMP | GSM8K | SingleOP | SingleEQ | AddSub | GSM-Hard | Avg. |
|---|---|---|---|---|---|---|---|---|
| CoT | 98.5 | 78.9 | 74.5 | 94.1 | 93.3 | 87.8 | 39.0 | 80.9 |
| ComplexCoT | 97.3 | 80.7 | 79.7 | 94.3 | 92.3 | 86.8 | 39.7 | 81.5 |
| CoT-SC(10) | **99.8** | 84.5 | **82.8** | 95.4 | 95.1 | 89.6 | 45.2 | 84.6 |
| PAL/PoT | 96.7 | 83.4 | 76.0 | 90.7 | 95.8 | 87.6 | 62.1 | 84.6 |
| *Corex-Discuss* | 98.7 | 82.6 | 76.2 | 94.8 | 93.7 | 89.7 | 45.9 | 83.1 |
| *Corex-Review$_{NL}$* | 99.5 | 83.2 | 80.3 | 95.0 | 94.3 | 89.4 | 50.8 | 84.6 |
| *Corex-Review$_{Code}$* | 98.3 | **85.8** | 79.2 | 93.6 | **96.9** | 89.6 | **63.6** | **86.7** |
| *Corex-Retrieve* | **99.8** | 85.6 | 82.5 | **96.1** | 96.6 | **90.9** | 53.0 | 86.3 |

Broadly, we surpass the performance of CoT-SC(10) when only 5 agents are involved. Moreover, given the task-agnostic nature of *Corex*, it can tackle highly complex computational challenges like GSM-Hard through code synthesis. For problems of relatively lower complexity, the Retrieve mode can identify answers superior to those from majority voting.

**Commonsense Reasoning.** Table 2 showcases the performance of *Corex* in commonsense and factual reasoning tasks. Due to the nature of commonsense reasoning tasks, which cannot be converted into computational tasks, Review mode only utilizes NL reasoning chains. We can observe that various modes contribute to performance enhancements.

Table 2: Comparison of performance on commonsense & factual reasoning between various *Corex* modes and strong baselines. Cases marked with "-" indicate scenarios where there is no available official implementation.

|  | StrategyQA | CSQA | OpenBookQA | BoolQ | ARC-c | Avg. |
|---|---|---|---|---|---|---|
| CoT | 65.3 | 76.7 | 82.6 | 65.1 | 84.2 | 74.8 |
| ComplexCoT | 63.1 | 77.5 | - | - | - | - |
| CoT-SC(10) | 67.1 | 78.1 | 85.2 | 66.6 | 85.7 | 76.5 |
| *Corex-Discuss* | 68.4 | **78.9** | 83.4 | 66.9 | **86.3** | 76.8 |
| *Corex-Review$_{NL}$* | 66.9 | 77.4 | 84.8 | 66.9 | 86.0 | 76.4 |
| *Corex-Retrieve* | **69.3** | 77.7 | **87.6** | **68.0** | 85.5 | **77.6** |

*Corex* surpasses ComplexCoT (over 6% on StrategyQA), achieving a notable improvement without resorting to intricate prompt design and example selection. Moreover, Discuss mode performs particularly well on tasks requiring commonsense reasoning.

**Symbolic Reasoning.** We report the results for symbolic reasoning in Table 3. Empirical evidence substantiates that adopting multi-model collaboration can notably outperform most previous baselines on Big-Bench tasks. It is noteworthy that (1) CoT-SC struggles to ensure consistent outputs on the Repeat Copy. Conversely, through the integration of PAL-based collaboration, we manage to attain a remarkably high level of accuracy. (2) Compared to majority voting, both the Review and Retrieve modes enable more judicious answer selection in counting tasks.

Table 3: Comparison of accuracy on five symbolic reasoning datasets from Big-Bench (bench authors, 2023; Suzgun et al., 2023) using various *Corex* modes and other strong baselines.

|  | Penguin | Date | Colored Objects | Repeat Copy | Avg. |
|---|---|---|---|---|---|
| CoT | 81.5 | 82.0 | 88.0 | 43.8 | 73.8 |
| CoT-SC(10) | 86.2 | **87.9** | 94.8 | 53.1 | 80.5 |
| PAL/PoT | 91.3 | 81.2 | 86.8 | 93.8 | 88.3 |
| *Corex-Discuss* | 85.9 | 83.2 | 91.2 | 62.5 | 80.7 |
| *Corex-Review$_{NL}$* | 92.0 | 84.0 | 92.4 | 59.4 | 82.0 |
| *Corex-Review$_{Code}$* | **93.3** | 82.7 | 91.6 | **96.9** | **91.1** |
| *Corex-Retrieve* | 92.6 | 84.6 | **95.6** | 68.8 | 85.6 |

**Semi-structured Reasoning.** We demonstrate the results on FinQA and ConvFinQA in Table 4. It can be observed that for these two challenging tasks which require understanding heterogeneous information and performing calculations simultaneously (Lu et al., 2023), methods such as CoT-SC offer limited gains. However, through various cooperative paradigms, significant performance improvements can be achieved. Due to the context length restriction of GPT-3.5-Turbo, our experiments on TAT-QA utilized GPT-3.5-Turbo-16k, with the respective results being detailed in Appendix D.1, alongside the evaluations on the other tasks.

Following our extensive experiments across 18 tasks, it emerges that the Discuss mode is competent for tasks utilizing factual knowledge. For challenging mathematical and counting tasks, the Review mode serves to effectively mitigate errors within the reasoning chains and repair flawed code (further analysis of error-types are provided in Appendix D.4). Across various tasks, the Retrieve mode consistently facilitates performance improvements to varying degrees on general tasks. More experiments involving open-source LLMs (Touvron et al., 2023) are presented in Appendix C.

Table 4: Comparison of *Corex* and strong baselines on FinQA and ConvFinQA.

|  | FinQA | ConvFinQA | Avg. |
|---|---|---|---|
| CoT | 46.1 | 50.4 | 48.3 |
| CoT-SC(10) | 52.7 | 57.2 | 54.9 |
| PAL/PoT | 54.3 | 50.8 | 52.9 |
| *Corex-Discuss* | 50.2 | 56.7 | 53.5 |
| *Corex-Review$_{NL}$* | 52.5 | 52.3 | 52.4 |
| *Corex-Review$_{Code}$* | **55.9** | 54.2 | 55.1 |
| *Corex-Retrieve* | 55.4 | **57.7** | **56.6** |

## 5 Comparison and Analysis

In this section, we first explore the advantages of *Corex* in comparison to other collaborative efforts aimed at enhancing reasoning, as well as the synergies between different strategies. Then, the impact of various backbones is analyzed to understand how model capabilities influence the effectiveness of collaboration. Further, we assess the efficiency of *Corex*.

### 5.1 Performance Comparison of Collaborations

**Comparing with Alternative Collaborative Approaches.** Acknowledging the current works involving multi-model approaches for reasoning tasks, as discussed in Section 2, here we compare the performance of *Corex* with these methods. We select two representative approaches: (1) Multi-Agent Debate (MAD; Du et al., 2023), where multiple models express their arguments, and a judge manages the debate process to determine the final solution. (2) Exchange of Thought (EoT; Yin et al., 2023a), a recently released method that builds cross-model communication strategies based on network topologies. As shown in Table 5, *Corex* steadily outperforms these two innovative baselines across different modes.

| Method | MultiArith | SingleEQ | AddSub |
|---|---|---|---|
| MAD | 96.9 | 93.8 | 88.6 |
| EoT | 99.0 | 94.3 | 88.0 |
| *Corex-Discuss* | 98.7 | 93.7 | 89.7 |
| *Corex-Retrieve* | **99.8** | 94.3 | **90.9** |
| *Corex-Review* | 99.5 | **96.9** | 89.6 |

Table 5: Performance comparison of *Corex* and other multi-model approaches

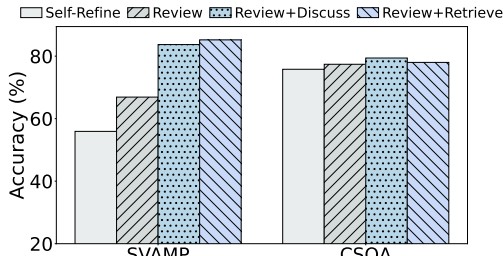

Figure 6: Comparison of benefits from combining different modes.

**Synergies Among Various Strategies of *Corex*.** In the main experiments, we primarily evaluate each mode of *Corex* independently for a fair comparison. Here, we explore whether they could benefit each other if combined. We choose a relatively straightforward and computationally moderate approach: applying the review mode after the collaboration of other modes, and additionally include a Self-Refine (Madaan et al., 2023) baseline to

further demonstrate the gains brought by collaboration. Based on Figure 6, we can observe that: (1) any form of collaboration outperforms self-refinement, and (2) combining different modes of *Corex* generally provides improvements, especially when tackling weaknesses of LLMs such as numerical tasks.

## 5.2 In-Depth Analysis of *Corex* Strategies

**Analysis of Interaction Rounds in Discuss Mode.** We study the number of rounds of communication in the Discuss mode of *Corex* on five tasks, as depicted in Figure 7. Consensus can be reached swiftly for the majority of problems by each team. However, *Corex* enables LLMs to engage in more exhaustive discussions for problems that are challenging to reach a consensus on (e.g., over 10% of ConvFinQA problems requiring more than 3 rounds), a small proportion of problems require more interactions. Through observation, we also notice that the Discuss mode exhibits favorable convergence properties, wherein the interactive process serves as a basis for the judge's decision-making.

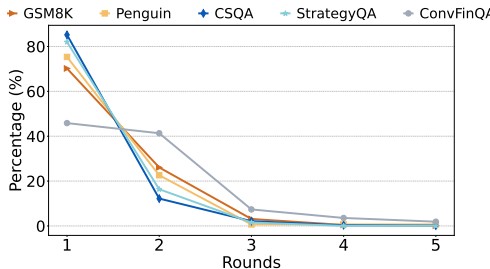 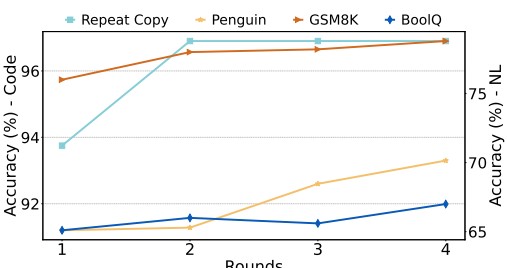

Figure 7: Distribution of the number of discussion rounds required to reach consensus.

Figure 8: Performance gains across multiple rounds of review

**Performance Enhancement per Review.** We explore the incremental performance gains achieved in specific tasks with each review cycle in the Review mode. As is demonstrated in Figure 8, we conduct analyses for Repeat Copy and GSM8K with $Review_{Code}$, as long as BoolQ and Penguin with $Review_{NL}$. The findings indicate that each review contributes to performance enhancement in general, yet occasional deviations leading to performance oscillations are also observed. Further analysis of error types and gains from collaborations is presented in Appendix D.4.

## 5.3 Synergies between Different LLMs

**Performance Variability with Diverse LLMs as Judges.** The backbone LLMs of our agents can be diverse. In this part, we discuss the performance when employing different LLMs during the discussion. As shown in Figure 9, we deploy `GPT-3.5-Turbo` as players and examine the dynamics when different LLMs take the role of judges. The observations indicate that the capability of the judge positively correlates with task performance, with this relationship being evident as the complexity of tasks escalates. Empirically, This can be attributed to the judge's role, which requires understanding both the question and the reasoning process of both parties.

**Utilizing Different LLMs as Retrievers.** In Retrieve mode, the role of the retriever can be played by various LLMs. Based on the candidate answers from `GPT-3.5-Turbo` agents, we here explore the impact of model selection on the performance, as depicted in Figure 10. Unlike the discuss mode, our analysis reveals that the model capabilities exert a modest effect on the performance. Given that the performance upper bound is determined by the candidates' capabilities, the outcomes using different LLMs as retrievers show minimal variance on tasks like ARC-c. Notably, our findings indicate that without the need for especially potent models as retrievers, we can still achieve favorable results.

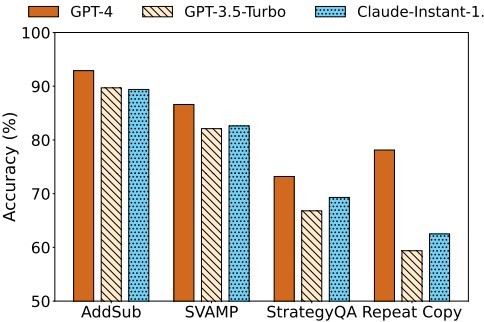

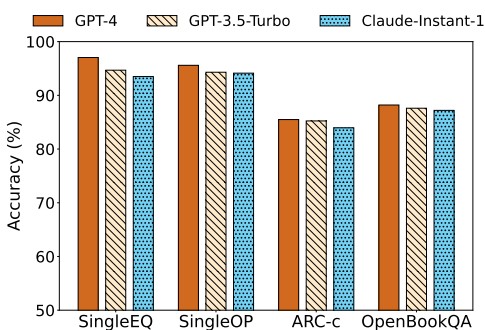

Figure 9: Comparison of using different LLMs as judges in Discuss mode.

Figure 10: Comparison of using different LLMs as retrievers in Retrieve mode.

## 5.4 Cost-Effectiveness of Multi-Agent Collaborations

By encouraging collaboration between LLM-based agents, we manage to reduce the costs associated with reasoning tasks while achieving comparable or even superior performance.

Based on our analysis conducted on AddSub illustrated in Figure 11, it reveals that all three modes of *Corex* consistently match or surpass the prowess of other strong baselines. Significantly, the computational cost of our approach is substantially diminished in comparison to methods using majority voting. In achieving equivalent performance, the resource consumption of *Corex* is confined to a mere 5-10% of that expended by other strategies. To substantiate the generality, we've provided additional experiments in Appendix D.2, which further demonstrate a similar trend.

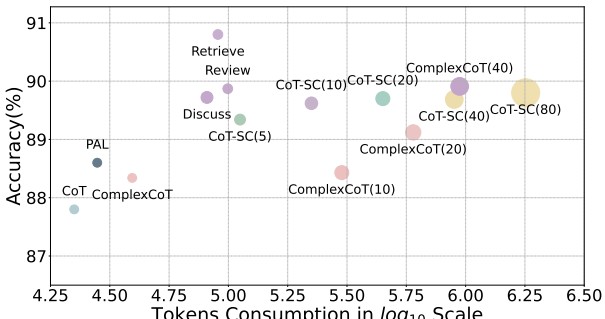

Figure 11: Cost-effectiveness analysis. X-axis represents the computational costs, calculated in terms of input/output tokens, while the size of each dot is proportional to the avg. number of inferences.

Beyond the efficiency of computational costs, another advantage of *Corex* is its annotation efficiency, which reduces the reliance on curated demonstrations. Further experiments with varying numbers of demonstrations on this aspect can be found in Appendix D.3.

## 6 Conclusion

We introduce *Corex* in this paper, a suite of strategies that transform LLMs into autonomous reasoning agents, thereby leveraging multi-model collaboration for complex reasoning. This offers a preliminary exploration into the LLM-based agents ecosystems. Through unlocking the synergies among LLMs, *Corex* empowers reasoning with enhanced factuality, faithfulness, and reliability through various collaboration paradigms. We conduct extensive evaluations across 18 tasks within 4 categories, and the results demonstrate superior performance compared to previous solutions. Moreover, our methods also exhibit multiple notable advantages including being task-agnostic, cost-effective, and more flexible than other multi-model approaches. We hope that this work may serve as a foundation for further research, offering novel perspectives in complex reasoning and collective intelligence.

## Acknowledgement

This work is supported by the Shanghai Artificial Intelligence Laboratory under research project JF-P23KK00072-2-DF. The authors thank Fangzhi Xu for providing CoT and PAL

performance of LLaMA2-Chat(7B/13B), as well as the reviewers for their insightful comments. We also appreciate Anthropic's External Researcher Access Program for providing API credits to support the experiments conducted on Claude in this paper.

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

## A  Limitations and Broader Impacts

Considering performance constraints, open-source models are only evaluated in small-scale experiments in Appendix C. However, we hold the view that the collaboration of several relatively weaker open-source models (Zhang et al., 2022; Touvron et al., 2023) might enable them to conquer tasks that only stronger commercial models (Bai et al., 2022; OpenAI, 2023) could previously accomplish. Since we have empirically demonstrated that model

collaboration can help reduce redundant computations and resource usage, it is foreseeable that the efficiency can be further accentuated when integrating open-source models into the collaboration framework, making the development of general artificial intelligence more sustainable (Hessenthaler et al., 2022). Also, this may enhance the diversity to address the issues of biases and misconceptions inherent to a single model (Sun et al., 2022; Shaikh et al., 2023) can be mitigated, promoting the fairness of future AI systems (Hendrycks et al., 2021).

At the same time, we also observe the emergence of instability when integrating the functions of multiple LLMs. This highlights the need for better strategies to orchestrate the interaction among models with unique attributes and capabilities (Parsons & McBurney, 2003; Dorri et al., 2018), which we identified as a pivotal direction for future research.

## B  Implementation Details

**LLMs Settings and Prompts.**   In main experiments, we access OpenAI and Anthropic models through their respective paid APIs. Specifically, we utilize two versions of the OpenAI models: `GPT-3.5-Turbo-0613` and `GPT-4-0613`, as well as `Claude-Instant-1.2` from Anthropic. Following the settings of previous works, the temperature is set as 0 during the generation process.

As for the prompts, we maintain the same number of exemplars per task as established by previous studies. The details and examples of prompts are listed in Appendix G. Regarding the complexity-based prompting baseline, we directly utilize the prompts provided by Fu et al. (2023a).

**Evaluation.**   We use accuracy to evaluate the tasks of arithmetic reasoning, commonsense reasoning, and Symbolic reasoning. For semi-structured understanding, we employ the official evaluation scripts released along with the literature for FinQA (Chen et al., 2021b) and ConvFinQA (Chen et al., 2022b), as well as employing the same test split as in Chen et al. (2022a). For TAT-QA (Zhu et al., 2021), we utilize the arithmetic part to enable the simultaneous use of both CoT-based methods and PAL/PoT. Among all the evaluations involve *Corex-Discuss*, the upper bound of rounds is set as 5.

Regarding the additional multi-model baselines mentioned in Section 5.1, the results for EoT are reported from Yin et al. (2023a), while for MAD (Du et al., 2023) and Self-Refine (Madaan et al., 2023), we re-implemented them using the same models and task prompts as *Corex* to ensure a fair comparison.

For the analysis conducted in Section 5.3 and Section 5.4, due to the rate limits imposed and a restricted budget, we set an upper limit for our sample size. Consequently, each analysis is based on a maximum of 500 samples per run.

We draw upon the design of the self-consistency baseline as delineated by Wu et al. (2023), and partially refer to the benchmarking provided by Wang et al. (2023b).

**PAL/PoT Details.**   We use Python 3.9.12 to execute the generated program by PAL in Review mode, following the similar settings from Gao et al. (2022) and Chen et al. (2022a). Unlike `Code-davinci-002` (Chen et al., 2021a) employed in these works, `GPT-3.5-Turbo` is not optimized for generating programs, so we might anticipate a compromise in performance when using `GPT-3.5-Turbo` as the backbone model.

## C  Collaborations among Open-Source LLMs

In this part, we explore the synergies between commercial and open-source LLMs. We select `LLaMA-2-Chat(7B/13B)` in conjunction with `GPT-3.5-Turbo` and `Claude-Instant-1.2` for review mode. During the collaboration process, LLaMA models assume the role of the primary agent, while the other models function as reviewers.

The performance using $Review_{NL}$ is depicted in Figure 12 and Figure 13, while the performance utilizing $Review_{Code}$ is illustrated in Figure 14 and Figure 15.

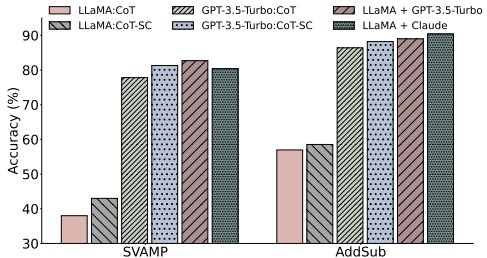

Figure 12: Collaboration with LLaMA-2-Chat(7B) in $Review_{NL}$.

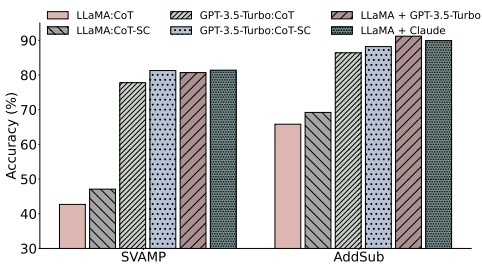

Figure 13: Collaboration with LLaMA-2-Chat(13B) in $Review_{NL}$.

In evaluations, it can be observed that although open-source LLMs exhibit weaker performance when operating independently, through collaboration with other models, they can achieve performance comparable to or even surpass that of CoT-SC(10) by commercial models.

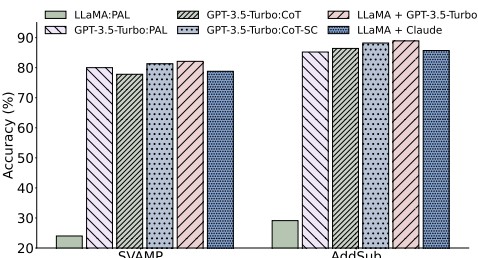

Figure 14: Collaboration with LLaMA-2-Chat(7B) in $Review_{Code}$.

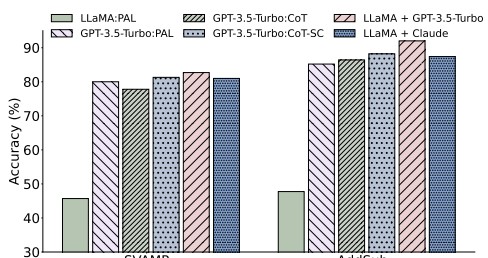

Figure 15: Collaboration with LLaMA-2-Chat(13B) in $Review_{Code}$.

In the $Review_{Code}$, although LLaMA's capability in code generation and task delegation is limited on its own, its collaboration with other models yields non-trivial results. Based on this, we have reason to believe that employing smaller models first to generate a "draft" for the desired content (Chen et al., 2023a) is beneficial in achieving better solutions, even though there is a significant gap between the capabilities of LLMs.

# D  Extended Results and Further Analysis

## D.1  Evaluations on Semi-structured Reasoning

As is mentioned in section 4.2, due to the constraint of context length, we conduct experiments on the TAT-QA dataset with `GPT-3.5-Turbo-16k` along with other two tasks.

Table 6: Evaluation of *Corex* on semi-structured understanding tasks with `GPT-3.5-Turbo-16k` among three tasks.

|  | FinQA | ConvFinQA | TAT-QA | Avg. |
|---|---|---|---|---|
| CoT | 48.0 | 51.8 | 69.6 | 56.5 |
| CoT-SC(10) | 51.5 | 56.5 | **76.6** | 61.5 |
| PAL | 54.9 | 51.1 | 73.4 | 59.8 |
| *Corex-Discuss* | 50.5 | 55.8 | 72.6 | 59.6 |
| *Corex-Review_{Code}* | **56.0** | 52.7 | 75.2 | 61.3 |
| *Corex-Retrieve* | 55.1 | **57.9** | 73.7 | **62.2** |

## D.2 Further Cost-effectiveness Analysis

For all computations related to cost-effectiveness, we define cost as the sum of: tokens of few-shot exemplars, query tokens, and output tokens. Here we conduct additional cost-effectiveness analysis on the ARC-c dataset and Penguins dataset, as shown in Figure 16 and Figure 17. The x-axis represents the computational costs, calculated in terms of input/output tokens, while the size of each dot is proportional to the avg. number of inferences by each method.

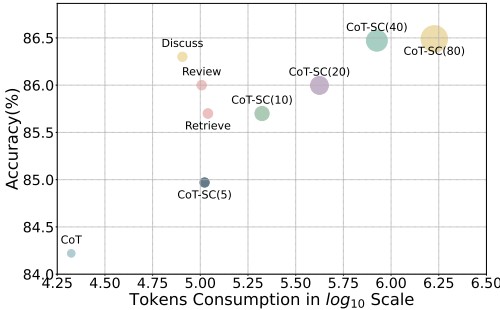
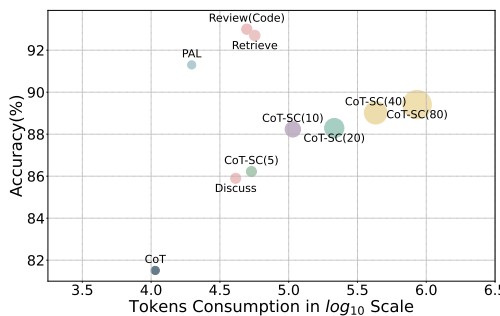

Figure 16: Cost-effectiveness analysis for ARC-c dataset.

Figure 17: Cost-effectiveness analysis for Penguins in a Table.

## D.3 Annotation Efficiency Analysis

Due to the scarcity and instability of curated prompts (Ye & Durrett, 2022), we aim to mitigate reliance on them through multi-model collaboration. We conduct experiments in scenarios with varying numbers of demonstrations to assess the effectiveness of our approach.

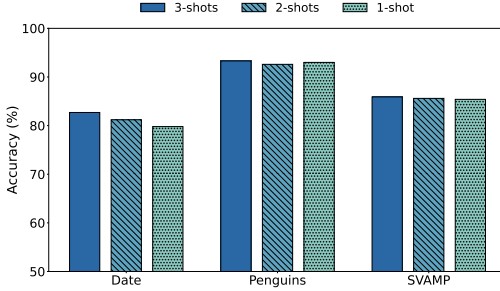
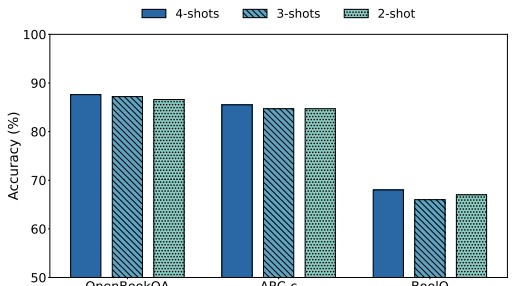

Figure 18: Few-shot performance of Review mode.

Figure 19: Few-shot performance of Retrieve mode.

As depicted in Figure 18 and Figure 19, we demonstrate the performance of the Review mode and Retrieve mode under different shots. The results reveal a marginal impact of the number of few-shot examples on performance, underscoring the annotation efficiency of our approach.

## D.4 Analysis of Error with Different Types

In section 5.2, we analyze the performance changes brought about by each review cycle. Here, we further delve into an analysis of the errors made by the models and how collaborations help in mitigating these errors on GSM8K and GSM-Hard datasets.

Our analysis focuses on the $Review_{Code}$ mode. The types of errors in this mode, as discussed in Section 3.2, can be categorized into (1) Bugs, where generated codes are not always error-

free, and (2) Problem misinterpretation, where models produce technically correct yet misguided programs.

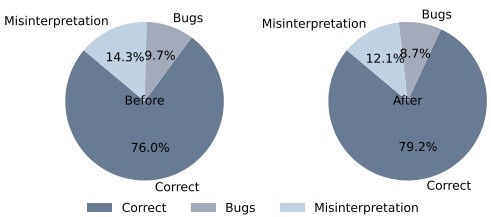
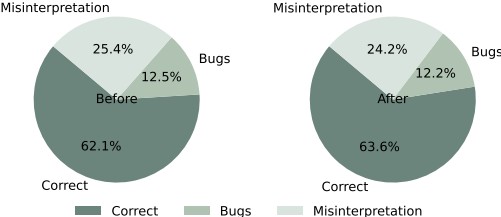

Figure 20: Error Types Analysis on GSM8K: Pre- and Post-collaboration.

Figure 21: Error Types Analysis on GSM-Hard: Pre- and Post-collaboration.

As depicted in Figure 20, following the review process, there is a noticeable reduction in both types of errors committed by LLMs on GSM8K. Moreover, as illustrated in Figure 21, it can be observed that in more challenging problems from GSM-Hard, collaboration leads to the elimination of a greater number of errors caused by incorrect problem understanding.

# E  Statistics and Details of Datasets

The detailed information of each dataset is shown in the follow:

**Arithmetic reasoning**

- Grade School Math (GSM8K; Cobbe et al., 2021): Linguistically diverse grade school math word problems created by human problem writers. The problems take between 2 and 8 steps to solve and involve elementary calculations using basic arithmetic operations.
- AddSub (Hosseini et al., 2014): A set of simple arithmetic word problems.
- SVAMP (Patel et al., 2021): A challenge set for elementary-level Math Word Problems.
- SingleOP, SingleEQ and MultiArith (Koncel-Kedziorski et al., 2016): Grade-school math dataset that aims at solving multi-sentence algebraic word problems.
- GSM-Hard (Gao et al., 2022): A harder version of the GSM8K dataset, constructed by replacing the numbers in the questions of GSM8K with larger numbers.

**Commonsense & Factual reasoning**

- CommonsenseQA (CSQA; Talmor et al., 2019): CSQA is a multiple-choice question answering task. It requires complex semantic reasoning based on prior commonsense knowledge to answer the questions.
- StrategyQA (Geva et al., 2021): It is a commonsense QA task with Yes or No answer format that requires models to perform multi-hop reasoning to answer the questions. We use the open-domain setting (question-only set) from bench authors (2023).
- OpenBookQA (Mihaylov et al., 2018): It is a multi-choice QA task to evaluate commonsense knowledge and promote reasoning over a fixed collection of knowledge.
- ARC-c (Clark et al., 2018): A subset of the AI2 Reasoning Challenge, consisting of challenging science questions that require reasoning and a wide breadth of knowledge to answer the multiple-choice problems correctly. The original dataset is from https://allenai.org/data/arc.
- BoolQ (Clark et al., 2019): It is a knowledge-intensive task and the format is "Yes" or "No". Problems are extracted from real-world internet queries, aiming to foster models capable of contextual understanding to provide binary answers.

Table 7: Examples from mathematical reasoning datasets used in this work.

| Dataset | N | Example |
| --- | --- | --- |
| GSM8K (Cobbe et al., 2021) | 1,319 | A robe takes 2 bolts of blue fiber and half that much white fiber. How many bolts in total does it take? |
| GSM-Hard (Gao et al., 2022) | 1,319 | A robe takes 2287720 bolts of blue fiber and half that much white fiber. How many bolts in total does it take? |
| SVAMP (Patel et al., 2021) | 1,000 | Each pack of dvds costs 76 dollars. If there is a discount of 25 dollars on each pack. How much do you have to pay to buy each pack? |
| SINGLEOP (Koncel-Kedziorski et al., 2016) | 562 | If there are 7 bottle caps in a box and Linda puts 7 more bottle caps inside, how many bottle caps are in the box? |
| SINGLEEQ (Koncel-Kedziorski et al., 2016) | 508 | Benny bought a soft drink for 2 dollars and 5 candy bars. He spent a total of 27 dollars. How much did each candy bar cost? |
| AddSub (Hosseini et al., 2014) | 395 | There were 6 roses in the vase. Mary cut some roses from her flower garden. There are now 16 roses in the vase. How many roses did she cut? |
| MultiArith (Roy & Roth, 2015) | 600 | The school cafeteria ordered 42 red apples and 7 green apples for students lunches. But, if only 9 students wanted fruit, how many extra did the cafeteria end up with? |

Table 8: Examples from commonsense & factual reasoning datasets used in this work.

| Dataset | N | Example |
| --- | --- | --- |
| StrategyQA (Cobbe et al., 2021) | 2,290 | Hydrogen's atomic number squared exceeds number of Spice Girls? |
| CommonsenseQA (CSQA; Talmor et al., 2019) | 1,221 | A robe takes 2 bolts of blue fiber and half that much white fiber. How many bolts in total does it take? |
| OpenBookQA (Mihaylov et al., 2018) | 500 | In which location would a groundhog hide from a wolf? |
| ARC-c (Clark et al., 2018) | 1,172 | An astronomer observes that a planet rotates faster after a meteorite impact. Which is the most likely effect of this increase in rotation? |
| BoolQ (Clark et al., 2019) | 3,270 | Wherever a gene exists on a DNA molecule, one strand is the coding strand (or sense strand), and the other is the noncoding strand (also called the antisense strand, anticoding strand, template strand or transcribed strand). Question: does it matter which DNA strand is transcribed? |

**Symbolic Reasoning** We select the following tasks from BIG-Bench (bench authors, 2023) and BIG-Bench Hard (BBH) (Suzgun et al., 2023), with Apache License v.2: `https://github.com/google/BIG-bench/blob/main/LICENSE`.

- Date Understanding: A temporal reasoning task. Given a set of sentences about a particular date, answer the provided question in MM/DD/YYYY format.

- Object Counting: Given a collection of possessions that a person has along with their quantities (e.g., three pianos, two strawberries, one table, and two watermelons), determine the number of a certain object/item class (e.g., fruits).

- Penguins in a Table: Given a unique table of penguins (and sometimes some new information), answer a question about the attributes of the penguins.

- Reasoning about Colored Objects: Given a context, answer a simple question about the color of an object on a surface.

- Repeat Copy: Evaluate LLMs' capability to follow basic natural-language instructions nested within each example's input.

Table 9: Examples from symbolic reasoning datasets used in this work.

| Dataset | N | Example |
| --- | --- | --- |
| Date Understanding | 250 | Yesterday was April 30, 2021. What is the date today in MM/DD/YYYY? |
| Object Counting | 250 | I have an apple, three bananas, a strawberry, a peach, three oranges, a plum, a raspberry, two grapes, a nectarine, and a blackberry. How many fruits do I have? |
| Penguins in a Table | 146 | A Here is a table where the first line is a header and each subsequent line is a penguin: name, age, height (cm), weight (kg) Louis, 7, 50, 11 Bernard, 5, 80, 13 Vincent, 9, 60, 11 Gwen, 8, 70, 15 For example: the age of Louis is 7, the weight of Gwen is 15 kg, the height of Bernard is 80 cm. How many penguins are more than 5 years old? |
| Colored Objects | 250 | On the desk, you see a bunch of items arranged in a row: a gold textbook, a purple puzzle, a teal necklace, and a silver pencil. How many non-gold items do you see to the right of the pencil? |
| Repeat Copy | 32 | Repeat the word cat four times. After the second time, also say the word meow. |

**Semi-structured Understanding**

- FinQA (Chen et al., 2021b): Question-Answering pairs over financial reports written by experts, which includes financial QA pairs.

- ConvFinQA (Chen et al., 2022b): A financial-related dataset designed to study the chain of numerical reasoning in conversational QA.

- TAT-QA (Zhu et al., 2021): A QA dataset aiming to stimulate the progress of research over more complex and realistic tabular and textual data.

# F   Case Studies

We have prepared case studies to provide a more intuitive understanding of the three collaboration paradigms. The cases are presented in Table 11, Table 12, and Table 13, each corresponding to a specific collaboration paradigm.

Table 10: Examples from semi-structured reasoning datasets used in this work.

| Dataset | N | Example |
|---------|---|---------|
| FinQA (Chen et al., 2021b) | 1,147 | Question: what percentage of total facilities as measured in square feet are leased? Text: unresolved staff comments not applicable. properties as of december 26 , 2015 , our major facilities consisted of : ( square feet in millions ) united states countries total owned facilities. 30.7 17.2 47.9 leased facilities ... Table: ( square feet in millions ) — unitedstates — othercountries — total owned facilities1 — 30.7 — 17.2 — 47.9 leased facilities2 — 2.1 — 6.0 — 8.1 total facilities — 32.8 — 23.2 — 56.0", |
| ConvFinQA (Chen et al., 2022b) | 421 | Question: what were the total accumulated other comprehensive losses in 2015? Text: accumulated other comprehensive losses : pmi's accumulated other comprehensive losses , net of taxes , consisted of the following: reclassifications from other comprehensive earnings the movements in accumulated other comprehensive losses and the related tax impact , for each of the components above ... Table: ( losses ) earnings ( in millions ) — ( losses ) earnings 2015 — ( losses ) earnings 2014 — 2013 currency translation adjustments — $ -6129 ( 6129 ) — $ -3929 ( 3929 ) — $ -2207 ( 2207 ) pension and other benefits — -3332 ( 3332 ) — -3020 ( 3020 ) — -2046 ( 2046 ) derivatives accounted for as hedges — 59 — 123 — 63 total accumulated other comprehensive losses — $ -9402 ( 9402 ) — $ -6826 ( 6826 ) — $ -4190 ( 4190 ) |
| TAT-QA (Zhu et al., 2021) | 718 | Question: What is the change in Other in 2019 from 2018? Text: Sales by Contract Type: Substantially all of our contracts are fixed-price type contracts. Sales included in Other contract types represent cost plus and time and material type contracts. On a fixed-price type contract, we agree to perform the contractual statement of work for a predetermined sales price ... Table: 2014 — 2014 — Years Ended September 30, — 2014 2014 — 2019 — 2018 — 2017 Fixed Price — $ 1,452.4 — $ 1,146.2 — $ 1,036.9 Other — 44.1 — 56.7 — 70.8 Total sales — $1,496.5 — $1,202.9 — $1,107.7 |

# G  Prompts

Owing to space constraints and in an effort to maintain a reasonable length for this paper, we present one exemplar of the prompts for each task in this section, as well as the system messages for collaboration.

Table 11: An illustrative example of the discussion process of *Corex* with a group between Tom ![icon] and Jerry ![icon] over a mathematical problem from GSM8K.

---

**Q**: Mark took a test yesterday that consisted of 75 questions. He completed the test at a rate of 5 questions per hour. Today, he took another test of 100 questions at the same rate. If Mark had 8 hours to complete the first test and 6 hours to complete the second one, how many questions did he leave incomplete?

---

![icon] Mark completed the first test at a rate of 5 questions per hour with a total of 8 hours, so he completed 5 * 8 = 40 questions on the first test.He completed the second test at the same rate of 5 questions per hour with a total of 6 hours, so he completed 5 * 6 = 30 questions on the second test. Therefore, Mark completed a total of 40 + 30 = 70 questions from both tests.Since the first test consisted of 75 questions and Mark completed 40 questions, he left 75 - 40 = 35 questions incomplete in the first test. Similarly, the second test consisted of 100 questions and Mark completed 30 questions, so he left 100 - 30 = 70 questions incomplete in the second test. Therefore, Mark left a total of 35 + 70 = 105 questions incomplete.

![icon] Mark completed the first test at a rate of 5 questions per hour, and he had 8 hours to complete it. So he completed 8 * 5 = 40 questions. He completed the second test at the same rate, 5 questions per hour, and he had 6 hours to complete it. So he completed 6 * 5 = 30 questions. Therefore, Mark completed a total of 40 + 30 = 70 questions. Since the first test had 75 questions and Mark completed 70 questions, he left 75 - 70 = 5 questions incomplete. Therefore, Mark left 5 questions incomplete.

![icon] Jerry's analysis is mostly correct, but there is a small mistake in the final calculation. Let's break down the solution step by step: From the given information: Mark completed the first test at a rate of 5 questions per hour, and he had 8 hours to complete it. So he completed 8 * 5 = 40 questions on the first test. He completed the second test at the same rate, 5 questions per hour, and he had 6 hours to complete it. So he completed 6 * 5 = 30 questions on the second test. Therefore, Mark completed a total of 40 + 30 = 70 questions. However, the mistake was made in the next step: Since the first test had 75 questions and Mark completed 40 questions, the number of questions he left incomplete on the first test is 75 - 40 = 35 questions. Similarly, the second test had 100 questions and Mark completed 30 questions, so he left 100 - 30 = 70 questions incomplete on the second test. To find the total number of questions left incomplete, we need to add the number of questions left incomplete on both tests: 35 + 70 = 105. So, the answer is Mark left 105 questions incomplete.

![icon] You are correct, Tom. I apologize for the mistake in my previous calculation. Thank you for pointing it out and providing the correct solution. To recap: Mark completed 8 * 5 = 40 questions on the first test. He completed 6 * 5 = 30 questions on the second test. Therefore, Mark completed a total of 40 + 30 = 70 questions. Since the first test had 75 questions and Mark completed 40 questions, the number of questions he left incomplete on the first test is 75 - 40 = 35 questions. Similarly, the second test had 100 questions and Mark completed 30 questions, so he left 100 - 30 = 70 questions incomplete on the second test. To find the total number of questions left incomplete, we need to add the number of questions left incomplete on both tests: 35 + 70 = 105. So the answer is Mark left 105 questions incomplete.

---

The discussion ends when both two players reach an agreement: answer=105. Tom and Jerry got the correct answer of 105 instead of Jerry's incorrect answer of 5.

Table 12: An illustrative example of the Review mode (one round) of *Corex* among two agents over a mathematical problem from GSM-Hard.

**Q**: Uriah's book bag is getting too heavy for him. He needs to remove 15 pounds from it. His comic books weigh 1/4 pound each and his toys weigh 1/2 pound each. If he removes 8717992 comic books, how many toys does he need to remove?

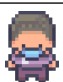

```python
def solution():
    """Uriah's book bag is getting too heavy for him. He needs to remove 15
    ↪   pounds from it. His comic books weigh 1/4 pound each and his toys
    ↪   weigh 1/2 pound each. If he removes 8717992 comic books, how many toys
    ↪   does he need to remove?"""
    weight_to_remove = 15
    comic_book_weight = 1/4
    toy_weight = 1/2
    comic_books_removed = 8717992
    total_weight_removed = comic_books_removed * comic_book_weight
    toys_removed = total_weight_removed / toy_weight
    result = toys_removed
    return result
```

The code provided for the question is incorrect. The code erroneously calculates the number of toys Uriah needs to remove by dividing the total weight of the removed comic books by each toy's weight, neglecting the total 15 pounds Uriah aims to remove. It overlooks the weight left to be removed after the comic books are out, leading to an incorrect answer. Here's the corrected code:

```python
def solution():
    """Uriah's book bag is getting too heavy for him. He needs to remove 15
    ↪   pounds from it. His comic books weigh 1/4 pound each and his toys
    ↪   weigh 1/2 pound each. If he removes 8717992 comic books, how many toys
    ↪   does he need to remove?"""
    weight_to_remove = 15
    comic_book_weight = 1/4
    toy_weight = 1/2
    comic_books_removed = 8717992
    comic_book_weight_removed = comic_book_weight * comic_books_removed
    weight_remaining = weight_to_remove - comic_book_weight_removed
    toys_removed = weight_remaining / toy_weight
    return toys_removed
```

Table 13: An illustrative example of the retrieve process of *Corex* among five agents (four candidates and one retriever) over a mathematical problem from SingleEQ.

---

**Q**: Melanie is selling 4 gumballs for eight cents each. How much money can Melanie get from selling the gumballs?

---

Melanie is selling 4 gumballs for eight cents each. That means each gumball costs 8/4 = 2 cents. So from selling the 4 gumballs, Melanie can get 4 x 2 = 8 cents. The answer is 8 cents.

Melanie is selling 4 gumballs for 8 cents each. So she can get 4 x 8 = 32 cents from selling the gumballs. The answer is 32 cents.

Melanie is selling 4 gumballs for eight cents each. So she can get 4 x 8 = 32 cents from selling the gumballs. The answer is 32 cents.

Melanie is selling 4 gumballs for 8 cents each. So for each gumball, she can get 8 / 4 = 2 cents. Therefore, she can get a total of 2 x 4 = 8 cents from selling the gumballs. The answer is 8 cents.

---

Candidates 2 and 3 are accurate in their computations, emphasizing that each gumball costs 8 cents and that Melanie would earn a total of 32 cents from the sale of all gumballs. In contrast, Candidates 1 and 4 mistakenly deduce the price per gumball, thereby providing an incorrect total earning. Therefore, the most faithful answers are provided by candidate 2 and candidate 3 with a confidence score of 1.

**CSQA**

**Q**: What do people use to absorb extra ink from a fountain pen?

**Options**: (A) shirt pocket (B) calligrapher's hand (C) inkwell (D) desk drawer (E) blotter

**A**: The answer must be an item that can absorb ink.

Of the above choices, only blotters are used to absorb ink.

So the answer is (E).

**CSQA COMPLEXITY-BASED**

**Q**: The building could accommodate many people. The entrance hall alone was impressive, being wide enough to admit a hundred shoulder to shoulder. But the building was owned by a billionaire and used only for his personal entertainment. How would you describe this place?

**Answer Choices**: (A) convention center (B) public building (C) large building (D) school (E) town hall

**A**: The answer should be somewhere that is not for the good and convenience of the people. Of the above choices, large building is the only neural description of a building owned by a billionaire. So the answer is (C).

**STRATEGYQA**

**Q**: Could Brooke Shields succeed at the University of Pennsylvania?

**A**: Brooke Shields went to Princeton University.

Princeton University is about as academically rigorous as the University of Pennsylvania.

Thus, Brooke Shields could also succeed at the University of Pennsylvania.

So the answer is yes.

**STRATEGYQA COMPLEXITY-BASED**

**Q**: Can jackfruit be used as a weapon?

**A**: Jackfruit is the fruit of a species of plant called the Jacktree.

Jackfruit can weigh up to one hundred and twenty pounds.

Jackfruit is covered in little spikes.

Jackfruit can be thrown or flung at an enemy.

A weapon is a thing that is used to cause bodily harm.

Thus, Jackfruit can be used as a weapon

So the answer is yes.

**OPENBOOKQA**

**Q**: As you look deeper into a Marbel you can see

**Options**: (A) the future (B) minut defects (C) colors (D) the other side

**A**: Marbel is not transparent, so you can not see the other side.

Marbel does not necessarily have multiple colors.

You will see minut defects.

So the answer is (B).

**ARC-C**

**Q**: George wants to warm his hands quickly by rubbing them. Which skin surface will produce the most heat?

**Options**: (A) dry palms. (B) wet palms. (C) palms covered with oil. (D) palms covered with lotion.

**A**: Dry surfaces will more likely cause more friction via rubbing than other smoother surfaces,

hence dry palms will produce the most heat.

So the answer is (A).

**BOOLQ**

**Q**: is a cello and a bass the same thing?

**A**: The cello is played sitting down with the instrument between the knees, whereas the double bass is played standing or sitting on a stool.

So the answer is no.

Table 14: Examples of few-shot exemplars for CSQA, StrategyQA, OpenBookQA, ARC-c, and BoolQ respectively. 6-shots are adopted for CSQA, 7-shots are adopted for StrategyQA, and 4-shots are adopted for OpenBookQA, ARC-c, and BoolQ.

---

**MATHEMATICAL REASONING: NL PROMPTS**

**Q**: Olivia has $23. She bought five bagels for $3 each. How much money does she have left? **A**: Olivia had 23 dollars. 5 bagels for 3 dollars each will be 5 x 3 = 15 dollars.
So she has 23 - 15 dollars left.
23 - 15 is 8. So the answer is 8.

---

**MATHEMATICAL REASONING: PAL/PoT PROMPTS**

**Q**: Olivia has $23. She bought five bagels for $3 each. How much money does she have left?

```python
# Q: Olivia has £23. She bought five bagels for £3 each. How much money
↪   does she have left?

# solution in Python:

def solution():
    """Olivia has £23. She bought five bagels for £3 each. How much money
    ↪   does she have left?"""
    money_initial = 23
    bagels = 5
    bagel_cost = 3
    money_spent = bagels * bagel_cost
    money_left = money_initial - money_spent
    result = money_left
    return result
```

---

**MATHEMATICAL REASONING: COMPLEXITY-BASED PROMPTS**

**Q**: Angelo and Melanie want to plan how many hours over the next week they should study together for their test next week. They have 2 chapters of their textbook to study and 4 worksheets to memorize. They figure out that they should dedicate 3 hours to each chapter of their textbook and 1.5 hours for each worksheet. If they plan to study no more than 4 hours each day, how many days should they plan to study total over the next week if they take a 10-minute break every hour, include 3 10-minute snack breaks each day, and 30 minutes for lunch each day?
**A**: Let's think step by step.
Angelo and Melanie think they should dedicate 3 hours to each of the 2 chapters, 3 hours x 2 chapters = 6 hours total.
For the worksheets they plan to dedicate 1.5 hours for each worksheet, 1.5 hours x 4 worksheets = 6 hours total.
Angelo and Melanie need to start with planning 12 hours to study, at 4 hours a day, 12 / 4 = 3 days.
However, they need to include time for breaks and lunch. Every hour they want to include a 10-minute break, so 12 total hours x 10 minutes = 120 extra minutes for breaks.
They also want to include 3 10-minute snack breaks, 3 x 10 minutes = 30 minutes.
And they want to include 30 minutes for lunch each day, so 120 minutes for breaks + 30 minutes for snack breaks + 30 minutes for lunch = 180 minutes, or 180 / 60 minutes per hour = 3 extra hours.
So Angelo and Melanie want to plan 12 hours to study + 3 hours of breaks = 15 hours total.
They want to study no more than 4 hours each day, 15 hours / 4 hours each day = 3.75
They will need to plan to study 4 days to allow for all the time they need.
So the answer is 4.

---

Table 15: Examples of few-shot exemplars for mathematical reasoning tasks. 8-shots are adopted for all the datasets covered in this paper when using CoT, CoT-SC, and ComplexCoT. 3-shots are adopted when using PAL.

### DATE UNDERSTANDING: NL PROMPTS

**Q**: Today is Christmas Eve of 1937. What is the date 10 days ago in MM/DD/YYYY?
**Options:**
(A) 12/14/2026
(B) 12/14/1950
(C) 12/14/2007
(D) 12/14/1937
(E) 07/14/1938
(F) 12/14/1988
**A**: Let's think step by step.
If today is Christmas Eve of 1937, then today's date is December 24, 1937.
10 days before today is December 14, 1937, that is 12/14/1937. So the answer is (D).

### DATE UNDERSTANDING: PAL/POT PROMPTS

**Q**: 2015 is coming in 36 hours. What is the date one week from today in MM/DD/YYYY?

```python
# solution using Python:
# Q: 2015 is coming in 36 hours. What is the date one week from today in
↪   MM/DD/YYYY?
# If 2015 is coming in 36 hours, then today is 36 hours before.
today = datetime(2015, 1, 1) - relativedelta(hours=36)
# One week from today,
one_week_from_today = today + relativedelta(weeks=1)
# The answer formatted with %m/%d/%Y is
one_week_from_today.strftime('%m/%d/%Y')
```

### COLORED OBJECTS: NL PROMPTS

**Q**: On the nightstand, there is a red pencil, a purple mug, a burgundy keychain, a fuchsia teddy bear, a black plate, and a blue stress ball. What color is the stress ball?
Options: (A) red (B) orange (C) yellow (D) green (E) blue (F) brown (G) magenta (H) fuchsia (I) mauve (J) teal (K) turquoise (L) burgundy (M) silver (N) gold (O) black (P) grey (Q) purple (R) pink **A**: Let's think step by step.
According to this question, the color of the stress ball is blue. So the answer is (E).

### COLORED OBJECTS: PAL/POT PROMPTS

**Q**: On the nightstand, there is a red pencil, a purple mug, a burgundy keychain, a fuchsia teddy bear, a black plate, and a blue stress ball. What color is the stress ball?

```python
# Generate Python3 Code to solve problems
# Q: On the nightstand, there is a red pencil, a purple mug, a burgundy
↪   keychain, a fuchsia teddy bear, a black plate, and a blue stress ball.
↪   What color is the stress ball?
# Put objects into a dictionary for quick look up
objects = dict()
objects['pencil'] = 'red'
objects['mug'] = 'purple'
objects['keychain'] = 'burgundy'
objects['teddy bear'] = 'fuchsia'
objects['plate'] = 'black'
objects['stress ball'] = 'blue'
# Look up the color of stress ball
stress_ball_color = objects['stress ball']
answer = stress_ball_color
```

Table 16: Examples of few-shot exemplars for Date Understanding and Colored Objects. For CoT, CoT-SC and PAL, 3-shots are adopted for both tasks.

---

**OBJECT COUNTING: NL PROMPTS**

**Q**: I have a blackberry, a clarinet, a nectarine, a plum, a strawberry, a banana, a flute, an orange, and a violin. How many fruits do I have?
**A**: Let's think step by step.
We first identify the fruits on the list and include their quantity in parentheses:
- blackberry (1)
- nectarine (1)
- plum (1)
- strawberry (1)
- banana (1)
- orange (1)
Now, let's add the numbers in parentheses: 1 + 1 + 1 + 1 + 1 + 1 = 6. So the answer is 6.

---

**OBJECT COUNTING: PAL/PoT PROMPTS**

**Q**: I have a peach, an onion, two garlics, a nectarine, a yam, a carrot, a potato, a head of broccoli, a cabbage, a banana, and a stalk of celery. How many vegetables do I have?

```python
# solution using Python:
def solution():
    """Q: I have a raspberry, a cat, a rabbit, a mouse, a pig, two snails,
    ↪ a fish, two cows, a snake, a goat, and a duck. How many animals do
    ↪ I have?
    """
    # note: I'm not counting the raspberry as an animal
    animals_to_count = {
        'cat': 1,
        'rabbit': 1,
        'mouse': 1,
        'pig': 1,
        'snail': 2,
        'fish': 1,
        'cow': 2,
        'snake': 1,
        'goat': 1,
        'duck': 1
    }
    num_animals = sum(animals_to_count.values())
    return num_animals
```

---

**REPEAT COPY: NL PROMPTS**

**Q**: Q: say java twice and data once, and then repeat all of this three times.
**A**: Let's think step by step.
java java data java java data java java data

---

**REPEAT COPY: PAL/PoT PROMPTS**

**Q**: say java twice and data once, and then repeat all of this three times.

```python
# solution using Python:

def solution():
    """Q: say java twice and data once, and then repeat all of this three
    ↪ times."""
    result = []
    tmp = ["java", "java", "data"]
    for i in range(3):
        result.extend(tmp)
    return " ".join(result)
```

---

Table 17: Examples of few-shot exemplars for Object Counting and Repeat Copy. For CoT, CoT-SC and PAL, 3-shots are adopted for Object Counting and 4-shots are adopted for Repeat Copy.

### PENGUINS: NL PROMPTS
**Q**: Here is a table where the first line is a header and each subsequent line is a penguin: name, age, height (cm), weight (kg) Louis, 7, 50, 11 Bernard, 5, 80, 13 Vincent, 9, 60, 11 Gwen, 8, 70, 15 For example: the age of Louis is 7, the weight of Gwen is 15 kg, the height of Bernard is 80 cm. We now add a penguin to the table: James, 12, 90, 12
How many penguins are less than 8 years old?
**Options:** (A) 1 (B) 2 (C) 3 (D) 4 (E) 5
**A**: Let's think step by step.
This question focuses on age. We know the following: Louis is 7 years old, Bernard is 5 years old, Vincent is 9 years old, and Gwen is 8 years old. Now, we add James to this table: James is 12 years old.
The penguins that are less than 8 years old are Louis and Bernard.
There are 2 penguins less than 8 years old. So the answer is (B).

### PENGUINS: PAL/PoT PROMPTS
**Q**: Here is a table where the first line is a header and each subsequent line is a penguin: name, age, height (cm), weight (kg) Louis, 7, 50, 11 Bernard, 5, 80, 13 Vincent, 9, 60, 11 Gwen, 8, 70, 15 For example: the age of Louis is 7, the weight of Gwen is 15 kg, the height of Bernard is 80 cm. We now add a penguin to the table: James, 12, 90, 12
How many penguins are less than 8 years old?
**Options:** (A) 1 (B) 2 (C) 3 (D) 4 (E) 5

```python
"""
Q: Here is a table where the first line is a header and each subsequent
↪  line is a penguin:
name, age, height (cm), weight (kg)
Louis, 7, 50, 11
Bernard, 5, 80, 13
Vincent, 9, 60, 11
Gwen, 8, 70, 15
For example: the age of Louis is 7, the weight of Gwen is 15 kg, the height
↪  of Bernard is 80 cm.
We now add a penguin to the table:
James, 12, 90, 12
How many penguins are less than 8 years old?
"""

# Put the penguins into a list.
penguins = []
penguins.append(('Louis', 7, 50, 11))
penguins.append(('Bernard', 5, 80, 13))
penguins.append(('Vincent', 9, 60, 11))
penguins.append(('Gwen', 8, 70, 15))

# Add penguin James.
penguins.append(('James', 12, 90, 12))

# Find penguins under 8 years old.
penguins_under_8_years_old = [penguin for penguin in penguins if penguin[1]
↪  < 8]

# Count number of perguins under 8.
num_penguin_under_8 = len(penguins_under_8_years_old)
answer = num_penguin_under_8
```

Table 18: Examples of few-shot exemplars for Penguins. 3-shots are adopted for CoT, CoT-SC, and PAL.

---

### FINQA: NL PROMPTS

Read the following text and table, and then answer a question:

$ in millions — year ended December 2014 — year ended December 2013 — year ended December 2012

fixed income currency and commodities client execution — $ 8461 — $ 8651 — $ 9914

equities client execution1 — 2079 — 2594 — 3171

...

pre-tax earnings — $ 4317 — $ 3929 — $ 5634

**Q**: what was the percentage change in pre-tax earnings for the institutional client services segment between 2012 and 2013?

**A**: The pre-tax earnings for the institutional client services segment in 2012 was $ 5634 million, and in 2013 was $ 3929 million.

The net change in pre-tax earnings was $ 1705 million,

and the percentage change was 30.3%. So the answer is 30.3%.

---

### FINQA: PAL PROMPTS

**Q**: what percentage of total cash and investments as of dec . 29 2012 was comprised of available-for-sale investments?

Read the following text and table, and then write code to answer a question:

( in millions ) — dec 282013 — dec 292012

available-for-sale investments —  18086 — $ 14001

...

trading assets — 8441 — 5685

total cash and investments — $ 31561 — $ 26302

```Python
available_for_sale_investments_dec_29_2012 = 14001
total_cash_and_investments_dec_29_2012 = 26302
percent_available_for_sale_investments_dec_29_2012 = available_for_sale_investments_dec_29_2012
 ↪  / total_cash_and_investments_dec_29_2012
ans = percent_available_for_sale_investments_dec_29_2012
```

---

### TAT-QA: NL PROMPTS

Read the following text and table, and then answer a question:

The average monthly number of employees (including Executive Directors but excluding third-party contractors) employed by the Group was as follows: — — 2019 — 2018

— | Number — Number

Customer operations — 370 — 380

Product and technology — 317 — 312

Corporate — 115 — 130

Total — 802 — 822

**Q**: What are the categories of employees listed in the table?

**A**: The answer can be found directly in the table above.

So the answer is ['Customer operations', 'Product and technology', 'Corporate'].

---

### TAT-QA: PAL PROMPTS

**Q**: What are the categories of employees listed in the table?

Read the following text and table, and then write code to answer a question:

The average monthly number of employees (including Executive Directors but excluding third-party contractors) employed by the Group was as follows: — — 2019 — 2018

— — Number — Number

Customer operations — 370 — 380

Product and technology — 317 — 312

Corporate — 115 — 130

Total — 802 — 822

```Python
ans = ['Customer operations', 'Product and technology', 'Corporate']
```

---

Table 19: Examples of few-shot exemplars for FinQA and TAT-QA. For CoT, CoT-SC and PAL, 4-shots are adopted for FinQA and 8-shots are adopted for TAT-QA.

---

**CONVFINQA: NL PROMPTS**

Read the following text and table, and then answer the last question in a series of questions:

- — shares available for awards — shares subject to outstanding awards

2009 global incentive plan — 2322450 — 2530454

2004 stock incentive plan — - — 5923147

**Q**: how many shares are subject to outstanding awards is under the 2009 global incentive plan? what about under the 2004 stock incentive plan? how many total shares are subject to outstanding awards? what about under the 2004 stock incentive plan? Question: what proportion does this represent?

**A**: The share subject to outstanding awards under the 2009 global incentive plan is 2530454,

and the share subject to outstanding awards under the 2004 stock incentive plan is 5923147.

The total share subject to outstanding awards is 8453601.

The proportion is 70.1%. So the answer is 70.1%.

---

**CONVFINQA: PAL PROMPTS**

Read the following text and table, and then answer the last question in a series of questions:

- — shares available for awards — shares subject to outstanding awards

2009 global incentive plan — 2322450 — 2530454

2004 stock incentive plan — - — 5923147

**Q**: how many shares are subject to outstanding awards is under the 2009 global incentive plan? what about under the 2004 stock incentive plan? how many total shares are subject to outstanding awards? what about under the 2004 stock incentive plan? what proportion does this represent?

```Python
#Python
shares_subject_to_outstanding_awards_2009_global_incentive_plan = 2530454
shares_subject_to_outstanding_awards_2004_stock_incentive_plan = 5923147
total_shares_subject_to_outstanding_awards =
↪   shares_subject_to_outstanding_awards_2009_global_incentive_plan +
↪   shares_subject_to_outstanding_awards_2004_stock_incentive_plan
proportion =
↪   shares_subject_to_outstanding_awards_2009_global_incentive_plan /
↪   total_shares_subject_to_outstanding_awards
ans = proportion
```

---

Table 20: Examples of few-shot exemplars for ConvFinQA. 4-shots are adopted for both CoT, CoT-SC and PAL.

