# OpenReview forum: "Corex: Pushing the Boundaries of Complex Reasoning through Multi-Model Collaboration"
_colmweb.org/COLM/2024/Conference — COLM_

### Official Review · Reviewer_SzzF · 2024-05-15

**Rating:** 8
**Confidence:** 5
**Ethics Flag:** 1

**Summary:**

The paper introduces Corex, aimed at enhancing the reasoning capabilities of Large Language Models (LLMs). Corex transforms LLMs into autonomous agents that collaborate through strategies inspired by human behavior, including Discuss, Review, and Retrieve modes.

Extensive experiments across diverse reasoning tasks demonstrate Corex's superior performance, highlighting its ability to overcome common errors and provide improved solutions.

The study emphasizes Corex's cost-effectiveness, the synergies between models of varying scales, and its contribution to annotation efficiency.

**Questions To Authors:**

- It's unclear how well the Discuss, Review and Retrieve paradigms would scale to slightly larger numbers of agents (e.g., 8 agents).

- Is there potential for combining different modes to further enhance performance? For instance, within the Discuss mode, teams could engage in reviewing each other's solutions (in Review mode), potentially leading to refined iterations of their initial proposals. This integration could leverage the strengths of both modes to achieve superior outcomes.

**Reasons To Accept:**

1. This paper introduces various collaboration schemes and incorporates the code interpreter tool, which are important aspects of multi-agent systems that have been largely overlooked or insufficiently discussed in previous literature.

2. The experiments are impressively extensive, covering 4 types of reasoning tasks and 18 datasets.

**Reasons To Reject:**

More multi-agent baselines can be added. For example, Du et al. (2023) and Liang et al. (2023) suggest two multi-agent debate frameworks with 3 agents, which bear similarities to your Discuss mode. From the comparison of Corex-discuss and these baselines, we can gain more insight about the impact of agent quantity on the system's performance.

References:

[1] Du et al. Improving Factuality and Reasoning in Language Models through Multiagent Debate. https://arxiv.org/abs/2305.14325

[2] Liang et al. Encouraging Divergent Thinking in Large Language Models through Multi-Agent Debate. https://arxiv.org/abs/2305.19118

---

> ### Author Rebuttal · Authors · 2024-05-27
>
> Deeply grateful for the review! Your positive comments are encouraging for us.
>
> > Regarding more multi-agent baselines / agent quantity
>
> We are aware that there are other multi-agent methods. In section 5.1, "Performance Comparison of Collaborations," we utilized the multi-agent debate frameworks you mentioned as a baseline, and the performance comparison with different Corex modes is presented in Table 5. The Discuss mode generally outperforms the multi-agent debate baseline, which we attribute to our group discussion mechanism.
>
> Regarding the number of agents, due to the design of Discuss, we require 2n+1 agents for the system. We believe the framework is generally scalable, but as the number of agents increases, the performance gains may increase but are likely to reach an upper bound (as shown in supplementary experiments in Appendix D.2). Choosing 5 agents is primarily to (1) enable fair comparisons with other methods (e.g., CoT/PAL methods) and (2) highlight the cost-effectiveness of Corex (e.g., outperforming CoT-SC{10,20...}).
>
> We will follow your suggestion to explore more multi-agent baselines as references in our revision.
>
> > Combining Different modes
>
> We have already considered combining different modes, and some relevant experiments are presented in Figure 5. We combined "reviewing each other's solutions" with "Discuss/Retrieve" and demonstrated the feasibility of combining them on mathematical and commonsense tasks to some extent.
>
> Due to page limits, we compressed this part of the experiments into analysis. In the revision, we will further discuss the integration of methods to explore the feasibility of combining different modes.
>
> Once again, thank you for your time and patience.

---

> > ### Comment · Reviewer_SzzF · 2024-06-03
> > **Reviewer Response**
> >
> > Thanks for your effort in clarifying my previous questions. I decide to maintain my original score which is already in favor of clear acceptance.

---

### Official Review · Reviewer_eqdr · 2024-05-23

**Rating:** 8
**Confidence:** 4
**Ethics Flag:** 1

**Summary:**

This paper introduces Corex, a suite of strategies that transform large language models into autonomous agents to enhance reasoning. Corex includes modes like Discuss, Review, and Retrieve, fostering collaboration among models. Through experiments across various reasoning tasks, Corex outperforms existing approaches, showcasing its effectiveness and potential for advancing artificial intelligence in natural language processing and complicated task-solving.

**Reasons To Accept:**

- The paper is well-written and easy-to-follow. The motivation is clear that the authors aim to leverage the interactions between multiple LLM agents to enhance the reasoning abilities. Good visualizations give readers a clear understanding of problem formulation, methodology, and empirical results.
- Some method design is novel, like the discussion between original agent with retrieval-augmented LLM agent.
- Abundant experiments across a set of different applications prove the validity of the proposed framework.

**Reasons To Reject:**

- Some method lacks novelty or lacks the authors' clarifications of difference compared with the existing methods. For example, what are the differences between Corex-discussion and multi-agent debate? What are the differences between Corex-review with self-reflection or self-debug?
- Similar to the above bullet, the paper also lacks comparison with additional baselines, including multi-agent debate, self-reflexion, self-debug, retrieval-augmented generation. These single model or single agent method might have better performance than multi-agent framework.
- The method relies heavily on the interactions between multiple LLM agents. From my perspective, the conclusions made from the discussion between LLMs are not always reliable, as LLM agents have the tendency to trust the other LLMs' suggestions, making it easy for them to change their own answers. Do the authors have some unique design to guarantee the quality of interactions?

---

> ### Author Rebuttal · Authors · 2024-05-27
>
> Thank you for your insightful review. Here we try to address each of the points you raised as follows:
>
> > Methodological: Corex-Discussion vs. Multi-Agent Debate
>
> As discussed at the end of *section 3.1*, current multi-agent debate approaches have several issues: (1) communication is limited by context length, (2) erroneous consensus or biases can arise, and (3) there is a risk of stronger LLMs monopolizing the collaboration. Therefore, we specifically designed the discussion mode to balance diversity and factuality in the collaboration process through group discussions + introduction of a judge.
>
> > Methodological: Corex-Review vs. Self-Reflection
>
> Compared to methods like self-reflection, Corex-Review offers continuous improvements in each round of review, whether for NL or code. This multi-agent approach helps mitigate issues like error accumulation and plateauing in text quality, as described in the beginning of *section 3.2*.
>
> > Empirical: Comparison with Additional Baselines
>
> Thank you for reviewing our experimental results! The baselines you mentioned have already been covered in *section 5.1*.
> 1. For methods like multi-agent debate, we selected MAD[1], EoT[2] as baselines. The experimental results are shown in *Table 5* demonstrate that Corex generally performs better in a multi-model scope.
> 2. For methods like self-reflection, we included self-refine in our experiments in *Figure 5* (due to page limits, this was compressed into the figure, which you might have missed).
>
> We will include more comparative methods in the references as you recommended.
>
> > Regarding LLM Agents' Interaction
>
> Precisely because the interactions between LLMs are not always reliable, we designed some novel mechanisms in Corex. For instance, in discussion, we maintain answer diversity through grouping and control the quality of interactions by introducing a judge. Regarding the point you raised about "making it easy for them to change their own answers," this is indeed a valuable area for further exploration. The existing multi-model/agent literature has not yet covered this aspect, and we will further discuss this to explore in the revision.
>
> Once again, thank you for your thorough review. We hope our response and the content provided in the paper can address your concerns!
>
> [1] Improving Factuality and Reasoning in Language Models through Multiagent Debate
>
> [2] Exchange-of-Thought: Enhancing Large Language Model Capabilities through Cross-Model Communication EMNLP'23

---

> > ### Comment · Reviewer_eqdr · 2024-05-27
> > **Reviewer Response**
> >
> > Thanks for your effort in responding to my previous questions. I have increased my score accordingly. Cheers and good luck!

---

### Official Review · Reviewer_Ysox · 2024-05-25

**Rating:** 6
**Confidence:** 4
**Ethics Flag:** 1

**Summary:**

This paper proposes a framework for improving LLM reasoning via multi-agent collaboration.
1. By prompting an existing LLM to take on different agent roles and have them collaborate in different patterns (i.e., Discuss, Review, and Retrieve), the paper shows empirically that it can lead to improved accuracy on 18 reasoning tasks among 4 categories.
2. The paper shows the effectiveness of their proposed agent collaboration strategy by comparing against existing model collaboration methods, and ablates the impact of using different LLMs for different agents in the collaboration.

**Questions To Authors:**

1. In terms of figure 8, please clarify the following:
    - I think “X-axis repre- sents the computational costs, calculated in terms of input/output tokens” is somewhat vague: is that the lump sum of the total input and output tokens for all the model calls during the course? If there’s a shared prefix among multiple calls, would that be discounted in the computation? Also token counts can only approximate the total computational cost: the computational cost for input/output tokens are different. It would be great if you can provide further breakdown in terms of the total input/output tokens.
    - What do you mean by “the size of each dot is proportional to the avg. number of inferences.”? Do you mean total # of calls the LLMs?
    - Can you provide a legend for the size of the bubbles?
2. Missing references and comparison, for example:
    - Self-Refine: Iterative Refinement with Self-Feedback
    - DSPy: Compiling Declarative Language Model Calls into Self-Improving Pipelines

**Reasons To Accept:**

1. This paper is nicely written; the method and results are clearly explained; the appendix properly documents the used prompts and experimental setup.
2. This idea is simple and easy to implement. It aims to use the “social model” to enable collaborations among LLM agents to improve the reasoning, which resembles the human collaboration process in the real world.
3. The experimental design is sound and there are sufficient experiments for measuring the performance of the proposed method; sufficient analysis is provided in both the main paper and appendix.

**Reasons To Reject:**

1. The novelty of the method contribution is not established. The key method of this paper is the agent collaboration approach, i.e., Discuss, Review, and Retrieve. These individual components are somewhat covered in the prior work (see below), and for most of the evaluation in the paper, these components are used separately (see from table 1 to 5). For example:
    - `Retrieve` seems to be similar to the self-consistency (SC) approach, in essence that both approaches sample multiple generations and coalesce the candidates with some mechanism. SC uses the model scoring while “Retrieve” uses models’ explicit judgment over the generations. It is also similar to DsPY’s demonstrate-and-search method.
    - `Review` reminds me of the self-refinement approach, whereas both of them iteratively use the LM to generate feedback to the generation and improve the output.
    - `Discuss` may feel like a combination of “longer” chain-of-thought plus self-refinement.
    - The authors claim to assign different personas (page 2) to the agents – that might be an interesting distinction from the proposed approach against existing methods – but that seems not to be the focus of the paper.
    - What are other potential patterns for collaboration and why are the three components chosen in this paper? For example, delegation is a common pattern in human collaboration and one could imagine using different agents to take on different parts of the task.
2. The empirical results are somewhat mixed. From table 1 to 5,
    - Often there is a baseline method that can achieve somewhat similar performance (within 1 or 2 points of absolute difference) to the proposed method. It would be helpful to provide standard errors in the evaluation so we can better understand if there’s a meaningful difference.
    - It’s unclear which of the collaboration strategies is optimal from the results. Sometimes there is high variance among the methods (see GSM-hard in table 1, Repeat Copy in table 3). And in such cases, the best proposed collaboration method achieves somewhat similar performance to one of the best performing baselines.
    - It would be interesting to see results on harder reasoning tasks like MATH, GPQA, as well as MMLU.
3. In terms of the scoping, I think this paper focuses a bit more on “multi-agent” collaboration than “multi-model” collaboration. If I understand correctly, in the main paper, the primary experiments are done via prompting one LLM for different agents, and the multi-model collaboration only happens at section 5.2 & 5.3, more of an ablation study rather than the main findings. If that’s the case, I suggest changing the “multi-model” in the title to “multi-agent” or change the presentation of the results for this paper.

---

> ### Author Rebuttal · Authors · 2024-05-27
>
> Thank you for your valuable suggestions!
>
> > Methodological Contribution
>
> Thank you for focusing on our method! Retrieve is designed specifically to address issues in SC, such as erroneous consensus. We will compare and discuss this in relation to the "demonstrate-and-search" method in the revision. Review offers continuous improvements in each round of review, whether for NL or code. This helps mitigate issues like error accumulation and text quality, as described at the beginning of Sec 3.2. Discuss mode is generally different from CoT+Self-refine; we innovatively *balance diversity and factuality* in LLM interactions through group discussions and the introduction of a judge.
>
> > Empirical Results
>
> Thanks for carefully examining our experiments. We have endeavored to cover a wide range of 18 datasets and have generally demonstrated the effectiveness of our method across strong baselines (Avg.). Due to the high costs, we were unable to perform experiments with error bars. For generalizability, we included experiments with different LLMs in Sec 5.2 and App C, which also confirmed the overall effectiveness.
> High variance in a few exps. is due to the nature of the datasets themselves. For example, the repeat copy dataset has only 32 cases (1 sample = +3.125 %), and GSM-Hard shows significantly better performance using code compared to NL.
>
> > Scoping
>
> Thanks for your suggestion. Indeed, our focus leans towards constructing multiple reasoning agents. We will follow your advice to adjust the presentation.
>
> > Fig 8
>
> The number of I/O tokens was calculated using tools like tiktoken, including the shared prefix (such as meta messages). Indeed, the costs of I/O tokens vary, and we will include an additional figure in the revision to provide a further breakdown, making the analysis clearer.
> "The size of each dot is proportional ...," Inferences referring to the total LLM calls. For example, for a single query: CoT/PAL = 1, CoT-SC(x) = x, Corex-Review = 5, etc. Due to limited space, we had to compress much information into one figure. We will include a legend in the revision. Thank you for helping us improve readability!
>
> > Missing Refs/Comparison
>
> We already included self-refine in Fig 5 (due to page limits, this was compressed into the figure, which you might have missed). We all agree that DSPy is promising for building reasoning agents, and we will follow your suggestion to include it in the references and discussions.
>
> Thank you again for your time and patience.

---

> > ### Comment · Reviewer_Ysox · 2024-06-06
> > **Thank you for the rebuttal!**
> >
> > Thanks for the rebuttal and clarification! I like this paper in general, but I share some similar concerns as LnBc:
> >
> > > From the main results in section 4, it seems that there’s no clear mode that wins all benchmarks within a bigger reasoning bucket (out of the 4 categories experimented). For a new benchmark or a new production usecase, it seems unclear which mode of Corex would a user of this method use and why.
> >
> > Would love to see your revision and detailed analysis!

---

> > > ### Author Response · Authors · 2024-06-07
> > > **Response from authors**
> > >
> > > Thank you for your response! As we mentioned in our reply to LnBc, the current version only provides a brief analysis due to page limits. In the revision, we will add more details, such as visualizing the "win rates" of different modes across various task types.
> > >
> > > Thank you again for your insightful suggestions.

---

### Official Review · Reviewer_LnBc · 2024-05-28

**Rating:** 6
**Confidence:** 3
**Ethics Flag:** 1

**Summary:**

This submission introduces Corex which aims to improve the reasoning of LLMs using multi-model collaboration. In order to solve complex reasoning tasks, Corex uses LLM-based agents collaborating with each other in different modes to find the best solution. The different modes are inspired by how humans work together and collaborate to solve problems.

In terms of quality, clarity, originality and significance of this work:
- The paper is well written and high quality. The experiments are thorough as they have investigated the performance of Corex on 18 different reasoning benchmarks and results seem to show that at least one of the Corex modes performs better than other baselines.
- In terms of clarity, the paper is overall clear, though there are parts which can be clarified  more (see section about questions below).
- The technical content is original, where the authors propose a new approach to improve complex reasoning capabilities of LLMs. The authors propose 3 different ways: Discuss, Review and Retrieve. Each of these methods have different motivations and attempts to address limitations of previous approaches.
- The method shows better/competitive performance in terms of number of tokens used which directly translate to reduced cost, which is an important consideration. There is also another angle of LLM agents engaged in multi-model collaboration which is an interesting and actively area of exploration.

**Questions To Authors:**

See suggestions / questions:

- [Typo] In section 1, “In this study, we propose Corex, a suite of human-inspired strategies that leveraging” -> leverages
- In section 1, “Additionally, Corex reduces the reasoning overheads, achieving multifaceted cost-effectiveness.”: What is the “multi-faceted” aspect?
- In several parts of the manuscript, the authors talk about “reasoning chain”. It would be great to provide a definition and/or example of that somewhere. I believe they are referring to the explanation of the answer of a given reasoning prompt?
- In section 3.1 when the authors talk about the “Discuss” mode, they mention:
```
We opt not to facilitate models in jointly exchanging their reasoning processes to converge on a single common answer for several reasons: (1) The limited context length inhibits the ability to hold the communication process, (2) A single final answer is not always correct, as erroneous consensus or biases among models can occur (Wang et al., 2023c), (3) Given performance gaps among various LLMs, there is a risk of strong ones “monopolizing” the collaborations, thereby overshadowing the insights from others
```
However, even with the green and blue team I'm not sure how the 3rd limitation is addressed. Moreover, since the experimentation has been around GPT-3.5-Turbo where information from only previous rounds is stored (from footnote 1), I believe limitation 1 above is also not handled?
- In the “Discuss” mode of Corex, have the authors experimented with k teams instead of 2?
- For the “Review” mode of Corex, have the authors considered also adding the execution signal of the code snippet in addition to a_p, c_p and m_p? It seems that they execute the final revised version. Maybe an intermediate signal of code execution would help?
- [Broader / aspirational] Have the authors thought about a generalizable framework where the choice among the 3 strategies of Corex can be task specific (and potentially agnostic to the user)?
- In section 3.3 which discusses the “Retrieve” mode: “...evaluates the faithfulness between ci and pi . Based on this assessment, the retriever assigns a” -> how?
- [Table 1] It is surprising to see code version performing worse than retrieve for some of the benchmarks.
- Do we have any intuition on why this is the case?
- Probably minor, but Figure 2 isn’t very clear. The change between R1 and R2 values isn't clear.
- I like figure 8. I wonder what the results look like for other benchmarks, maybe not for all 18 benchmarks, but at least one of each 4 categories?

**Reasons To Accept:**

1. The paper presents a new approach Corex that leverages multi-model collaboration to address complex reasoning tasks. The approach has 3 different modes and each of the modes attempts to address limitations of single LLMs. The modes are human-inspired and demonstrate competitive results for different benchmarks investigated.

2. The authors do a great job with comprehensive benchmarking across 4 different categories of reasoning tasks and 18 different actual tasks. The data and prompts are also made available for each of the benchmarks.

3. The angle of cost effectiveness seems promising to me (for e.g. figure 8 and more in appendix). Corex seems to outperform other iterative strategies at least for some of the tasks.

**Reasons To Reject:**

1. From the main results in section 4, it seems that there’s no clear mode that wins all benchmarks within a bigger reasoning bucket (out of the 4 categories experimented). For a new benchmark or a new production usecase, it seems unclear which mode of Corex would a user of this method use and why.

2. I think there’s somewhat unclear intuition on why the different modes are working well and justification for the exact setup for a given mode. For instance, I wonder why “discuss” mode has only 2 teams, green and blue? Did the authors experiment / think about k groups?  Same for “review” mode, rationale for one agent to be a primary one seems arbitrary.

3. [Minor] Beyond the results from Appendix section C, it seems to be that there’s missing experimentation around open source models. Adding results from open source LLMs like Llama3, Mistral, etc would be even more informative for the broader community.

---

> ### Author Rebuttal · Authors · 2024-05-29
>
> Thank you for reviewing our paper!
> > Which Mode to Choose
>
> Thanks for closely examining the exps. Due to page limits, we only provided brief analysis of the performance of different modes across the table captions and at the end of page 7. We will discuss this in depth in the revision.
>
> > Intuition of Modes
>
> The 2 teams design in "discuss" primarily aims to allow the LLMs to think separately during interactions, which can be extended to any number of teams. As per your suggestion, we will include more discussion/results. "Review" has variants worth exploring, such as combining voting. Since we compared this to self-refine, we focused more on enhancing the capabilities through reviewing.
>
> > More Open-Source Exps.
>
> As you noted, current exps. are based on Llama2, whose performance is relatively weak. Since Llama3 was released after our submission, we plan to update the open-source backbones in the revision.
>
> > Reasoning Chain
>
> Apologies. Indeed, "explanations of the answer" should be clarified. We will fix this to make the paper more self-contained.
>
> > Multi-faceted Effectiveness
>
> The manuscript shows the effectiveness of token consumption. Other aspects such as Annotation Efficiency are presented in D.4.
>
> > Limitation of Perf. Gaps
>
> We conducted exps. with judges of different capabilities (Fig 6). However, studying the performance gaps between team agents is challenging due to context length and inherent ability. We will expand on this in discussion.
>
> > Num of Agents
>
> Due to the design of Discuss, we require 2n+1 agents. Here we choose to balance perf. and cost, and will include more in the revision.
>
> > Review+Exec.
>
> We think this is a great idea, but it will allow multiple attempts that might indirectly compromise the fairness of exps. However, incorporating exec. feedback when solving real-world problems is promising.
>
> > Task-Specific/Agnostic
>
> Brief analyses are provided at the end of page 7. We will follow your idea to include more cases/insights and further discuss the design of generalizable multi-model mechanisms.
>
> > Retriever
>
> We think that selecting is simpler than generating, while faithfulness can be a focus specified in the prompt.
>
> > w/o Code
>
> We observe that code-based solutions have issues like exec. errors (D.5). Sometimes selecting based on the model's reasoning process can be more reliable.
>
> > Fig 8
>
> Similar exps. on other benchs are provided in D.3.
>
> Thanks again, we will fix the typos and make further revisions (including Fig 2).

---

> > ### Comment · Reviewer_LnBc · 2024-06-01
> >
> > Thanks for your response.
> >
> > I think i didn't get the answer to my question earlier:
> > > In section 3.3 which discusses the “Retrieve” mode: “...evaluates the faithfulness between ci and pi . Based on this assessment, the retriever assigns a” -> how?
> >
> > Re: Similar exps. on other benchs are provided in D.3.
> > Those seem to be not for all the 4 categories right?
> >
> > Re: We think this is a great idea, but it will allow multiple attempts that might indirectly compromise the fairness of exps.
> > I don't think there are any "multiple attempts" in here. I'm basically saying that the execution signal / error would help the overall review process.

---

> > > ### Author Response · Authors · 2024-06-01
> > > **Further responses from the authors**
> > >
> > > Thank you very much for your reply! We apologize for any confusion caused by compressing our answers due to the rebuttal's character limit.
> > >
> > > > Evaluate the faithfulness
> > >
> > > Sorry for not explaining this clearly in the text. "Assign" is a process of LLM-based evaluation. The Retriever is instructed to evaluate candidates' answers by considering the faithfulness between the reasoning process and their answers. Here, we use definition of faithfulness: “if the reasoning process derived from the model can accurately be expressed by an explanation, we call it faithful.”[1,2]
> > >
> > > It is essentially an LLM evaluating the generated content of other LLMs. We will further clarify these details in the revision.
> > >
> > > [1] Faithful Chain-of-Thought Reasoning IJCNLP-AACL 2023
> > >
> > > [2] Towards Faithfully Interpretable NLP Systems: How Should We Define and Evaluate Faithfulness? ACL 2020
> > >
> > >
> > > > Similar Cost-effective Experiments on Other Benchmarks
> > >
> > > Yes, the main paper presents AddSub as a mathematical task. In Appendix D.3, we presented results for ARC-c and Penguins, which correspond to commonsense and symbolic tasks, respectively. Due to the high testing costs for TableQA-type tasks, we currently have not conducted those experiments. We will try to include these in the revision.
> > >
> > > > "Multiple Attempts"
> > >
> > > We agree that execution signals and error feedback would improve the overall review process. In real-world scenarios, this would enhance the effectiveness of the entire system. In the rebuttal, our reference to "multiple attempts" means that in general reasoning tasks, errors such as (I) code execution failures or (II) code execution correctness but incorrect answers are both counted as errors. Therefore, for evaluating reasoning tasks, if execution feedback is provided in each round, it validates whether type I errors occur in every review. Compared to methods like PAL/PoT, incorporating execution signals provides additional "verification" opportunities.
> > >
> > > Once again, thank you for your feedback!

---

> > > > ### Comment · Reviewer_LnBc · 2024-06-03
> > > >
> > > > Thanks for clarifying and taking the time to respond!

---

### Decision · Program_Chairs · 2024-07-10

**Decision:**

Accept

**Comment:**

Summary:
The paper presents Corex, which targets the improvement of Large Language Models' (LLMs) reasoning abilities by turning them into collaborative autonomous agents using human-inspired strategies like Discuss, Review, and Retrieve modes.
Overall the authors are willing to make adjustments to presentations about multi-agent instead of multi-model, add more open-source model experiments, and parallels between this approach to self-consistency, self-refine, CoT etc.,.

Pros:
1. The paper introduces Corex, a novel method that utilizes multimodal collaboration to tackle complex reasoning tasks through three distinct, human-inspired modes, each designed to overcome the limitations of individual LLMs and achieve competitive results across various benchmarks.
2. Thorough experimentation on 18 different reasoning tasks demonstrating that one of the modes from the proposed Corex model performs better than competitive baselines at least. The authors made the data and the benchmark available.
3. I believe that the cost-effectiveness aspect appears promising.

Cons:
1. The intuition behind why the different modes are effective and the justification for their specific setups seem somewhat unclear. It appears that no single mode consistently outperforms all benchmarks within the broader reasoning category, thereby making it unintuitive as to which Corex mode to choose for a new task. This is not adequately answered in the rebuttal directly but the authors promise to add relevant information in the paper.
2. There seems to be minimal exploration with open-source models. Incorporating outcomes from open-source LLMs such as Llama3 and Mistral could yield richer insights for the wider community. (as also pointed out by the reviewer LnBc). The authors promise to add these results in the paper.

Originality:
1. The authors offer a novel technique to boost the complex reasoning skills of LLMs. They outline three separate methods: Discuss, Review, and Retrieve, each driven by different motivations and designed to tackle the weaknesses of existing methods.
2. Feedback about a discussion/empirical comparison with SC, SR, CoT would differentiate this work from prior work for originality

Significance:
1. The proposed Corex method demonstrates improved or competitive performance by using fewer tokens, leading to reduced costs.
2. This work has a strong potential impact on the active area of research involving LLM agents collaborating in multi-model settings.

[comments from PCs] Please follow up on the AC recommendations, especially as far as exploring open-weight/open-source models.